# Underpinning heterogeneity in synaptic transmission by presynaptic ensembles of distinct morphological modules

Adam Fekete[1,2], Yukihiro Nakamura [3], Yi-Mei Yang[1,2,4], Stefan Herlitze[5], Melanie D. Mark[5], David A. DiGregorio[6,7] & Lu-Yang Wang[1,2]

Synaptic heterogeneity is widely observed but its underpinnings remain elusive. We addressed this issue using mature calyx of Held synapses whose numbers of bouton-like swellings on stalks of the nerve terminals inversely correlate with release probability (Pr). We examined presynaptic $Ca^{2+}$ currents and transients, topology of fluorescently tagged knock-in $Ca^{2+}$ channels, and $Ca^{2+}$ channel-synaptic vesicle (SV) coupling distance using $Ca^{2+}$ chelator and inhibitor of septin cytomatrix in morphologically diverse synapses. We found that larger clusters of $Ca^{2+}$ channels with tighter coupling distance to SVs elevate Pr in stalks, while smaller clusters with looser coupling distance lower Pr in swellings. Septin is a molecular determinant of the differences in coupling distance. Supported by numerical simulations, we propose that varying the ensemble of two morphological modules containing distinct $Ca^{2+}$ channel-SV topographies diversifies Pr in the terminal, thereby establishing a morpho-functional continuum that expands the coding capacity within a single synapse population.

[1] Program in Neurosciences and Mental Health, The Hospital for Sick Children, 555 University Ave, Toronto, ON M5G 1X8, Canada. [2] Department of Physiology, University of Toronto, Toronto, ON M5S 1A8, Canada. [3] Department of Pharmacology, Jikei University School of Medicine, Nishishinbashi, Minato-ku, Tokyo 1058461, Japan. [4] Department of Biomedical Sciences, University of Minnesota Medical School, 1035 University Drive, Duluth, MN 55812, USA. [5] Department of General Zoology and Neurobiology, Ruhr-University Bochum, Universitätsstrasse 150, D-44780 Bochum, Germany. [6] Unit of Dynamic Neuronal Imaging, Institut Pasteur, 25 rue du Dr Roux, 75724 Paris Cedex 15, France. [7] Centre National de la Recherche Scientifique (CNRS), UMR 3571, Genes, Synapses and Cognition, Institut Pasteur, 25 rue du Dr Roux, 75724 Paris Cedex 15, France. Correspondence and requests for materials should be addressed to L.-Y.W. (email: luyang.wang@utoronto.ca)

Strength and short-term plasticity (STP) are diverse across synapses[1]. Functional heterogeneity was described for many scenarios, even for single population of synapses between anatomically defined cell types, including autapses[2–6]. Release probability (Pr) of nerve terminals is considered the main parameter in diversifying synaptic strength and the polarity of STP, ranging from facilitation to depression and a mixture of the two[7,8], creating distinct operational modalities. Diversity of STP provides computational potential, e.g. different frequency filtering properties[9] that enable neural circuits to perform feature extraction[10]. Functional synaptic diversity can contribute to temporal coding of specific input modalities as well as enhancing pattern decorrelation[3,11,12].

Single population of synapses can vary in shape and size of the pre- and postsynaptic side[4,13,14], but the key molecular determinants and functional implications are still elusive[15], particularly for the presynaptic terminals. We previously explored the morphological variability of the mature calyx of Held synapse (≥P16[4]), a giant glutamatergic terminal in the auditory brainstem capable of high-fidelity and ultrafast neurotransmission for preserving timing and intensity cues critical for sound localization[16–18]. We found that mature calyces are composed of different proportions of two morphological modules, the thick digit-like stalks and the small bouton-like varicosities, called swellings. The swellings are connected to stalks through narrow and short neck, and contain SV assemblies and multiple active zones (AZ; [4,19,20]). We defined calyx complexity by the number of swellings, which vary across the population even after the sensitive period of auditory development[4,21]. Heterogeneity in the number of swellings on stalks has been confirmed in vivo[22]. We found that structural complexity is a strong predictor of synaptic function, including Pr, number of available SVs (readily releasable pool, RRP), STP, and fidelity of postsynaptic spiking[4], indicating that morphological variability supports functional diversity. However, the underlying mechanisms of this morpho-functional continuum have not been identified.

Strength and precision of synaptic transmission is influenced by the number of voltage-gated $Ca^{2+}$ channels (VGCCs) clustered in the active zone (AZ) and the proximity of synaptic vesicles (SV) to VGCCs[2,14,23–28]. A recent ultrastructural analysis revealed that the Pr and the number of presynaptic VGCCs scale with the AZ area, providing morphological correlate of the diversity in synaptic strength[14]. Recordings of $Ca^{2+}$ current, SV release and Pr at single AZs of immature calyx have shown that the number of VGCCs in clusters determines Pr and number of release-ready SVs, resulting in heterogeneous release properties among different AZs[26]. Freeze fracture replica labeling (SDS-FRL) of VGCCs revealed a clustered topographical arrangement that drives SV fusion from its periphery, and the distance between cluster and SV can account for developmental changes in synaptic transmission[27]. However, whether and how variations in this topography generate diversity in synaptic function within a single synapse population remains unknown. Here, we demonstrate that the global Pr of any given calyx is dictated by different proportions of two distinct morphological modules each with differing functional properties. High Pr stalk modules contain large VGCC clusters tightly coupled to SVs, while low Pr swellings modules employ more loosely coupled small VGCC clusters. By increasing the number of low Pr modules, the fidelity and sustainability of neurotransmission increases as a result of an expanded RRP size.

## Results

### Synaptic heterogeneity scales with morphological complexity.
We previously discovered a morphological correlate for functional diversity at mature calyces: increasing the number of swellings on the terminal results in a lower whole-terminal Pr while at the same time increases the reliability of high-frequency postsynaptic spiking during long trains[4]. To gain insights into how heterogeneity in the number of swellings influences heterogeneity in synaptic function, we investigated synaptic strength, quantal parameters, and STP with two morphological extremes: simple calyces with ≤10 swellings versus complex calyces with >20 swellings (Fig. 1a). When we stimulated the afferent axon using a bipolar electrode (Fig. 1a; 300 Hz, 200 ms), the amplitude of the first excitatory postsynaptic current (EPSC) was larger in swelling-rich complex than simple calyces (Fig. 1a, b). Since synaptic strength is determined by Pr and the size of RRP, we calculated the size of RRP by linear back-extrapolation from the last 50 ms of the steady-state part of the cumulative EPSC curve to 0 ms of the action potential (AP) train (y-axis intercept; [29]), and the Pr from the ratio of EPSC1 amplitude over RRP size. We found that complex calyces have smaller initial Pr than the simple ones, but contain significantly larger RRP (Fig. 1b). An alternative method[30] yielded slightly increased RRP size and decreased Pr, however, the heterogeneity among calyces is comparable with the two methods (Fig. 1b, dots next to bar graphs).

Since initial Pr influences STP during the train[7,8], we quantified the EPSC paired-pulse ratios (PPR, ratio of EPSC2 over EPSC1). We found that the PPR was smaller in simple than complex calyces, as predicted by their initial Pr (Supplementary Figure 1a). We examined the time course of short term depression (STD) in mature calyces by fitting the decay of EPSC train with a double exponential function. The fast and slow decay time constants reflect two subpools of SVs, a fast- and a slowly releasing one. The decay time constants in the two subpools and the size of the fast releasing subpool (A1) were not different between simple and complex calyces (Supplementary Figure 1b,c), but the size of the slowly releasing subpool (A2) was significantly larger in complex than simple calyces (Supplementary Figure 1c). We noted the slight deviation of EPSC amplitudes at steady-state depression from a straight line, likely reflecting a $Ca^{2+}$-dependent augmentation of SV replenishment (Fig. 1a; [31]) rather than a rebound from desensitization or saturation because desensitization or saturation are absent in mature calyces whereas the depletion of RRP is the main mechanism of STD[32–37]. To exclude postsynaptic mechanisms of heterogeneity we quantified the miniature EPSCs (mEPSC) in simple and complex calyces. We found that the amplitude and time course was identical in the two groups suggesting no major differences in the composition or gating of glutamate receptors (Supplementary Figure 2a,b). The correlation between the quantal parameters and the number of swellings indicates that the synaptic heterogeneity resides in presynaptic loci (Fig. 1b). Since the EPSC amplitude is the product of Pr (Fig. 1b), number of release sites (N), and quantal size (Supplementary Figure 2), we calculated that the number of release sites is indeed 55% larger for complex ($N = 1247$) than simple calyces ($N = 806$). These results indicate that swellings expand the morphological reservoir for harboring additional SVs in the RRP for release with low Pr.

### Action potential waveform is preserved throughout the calyx.
The fidelity of spike waveform and propagation into different compartments controls the magnitude of presynaptic $Ca^{2+}$ entry[38] and thus can potentially contribute to differences in Pr between simple and complex calyces. We performed sequential or simultaneous (dual) cell-attached recordings of compound action potential current (pre-$I_{AP}$) from swellings and stalks of the same calyx, while the afferent axon was stimulated using bipolar electrode (Fig. 2a). In some cases, we were able to patch the same calyx at multiple locations. Wherever we were able to register pre-

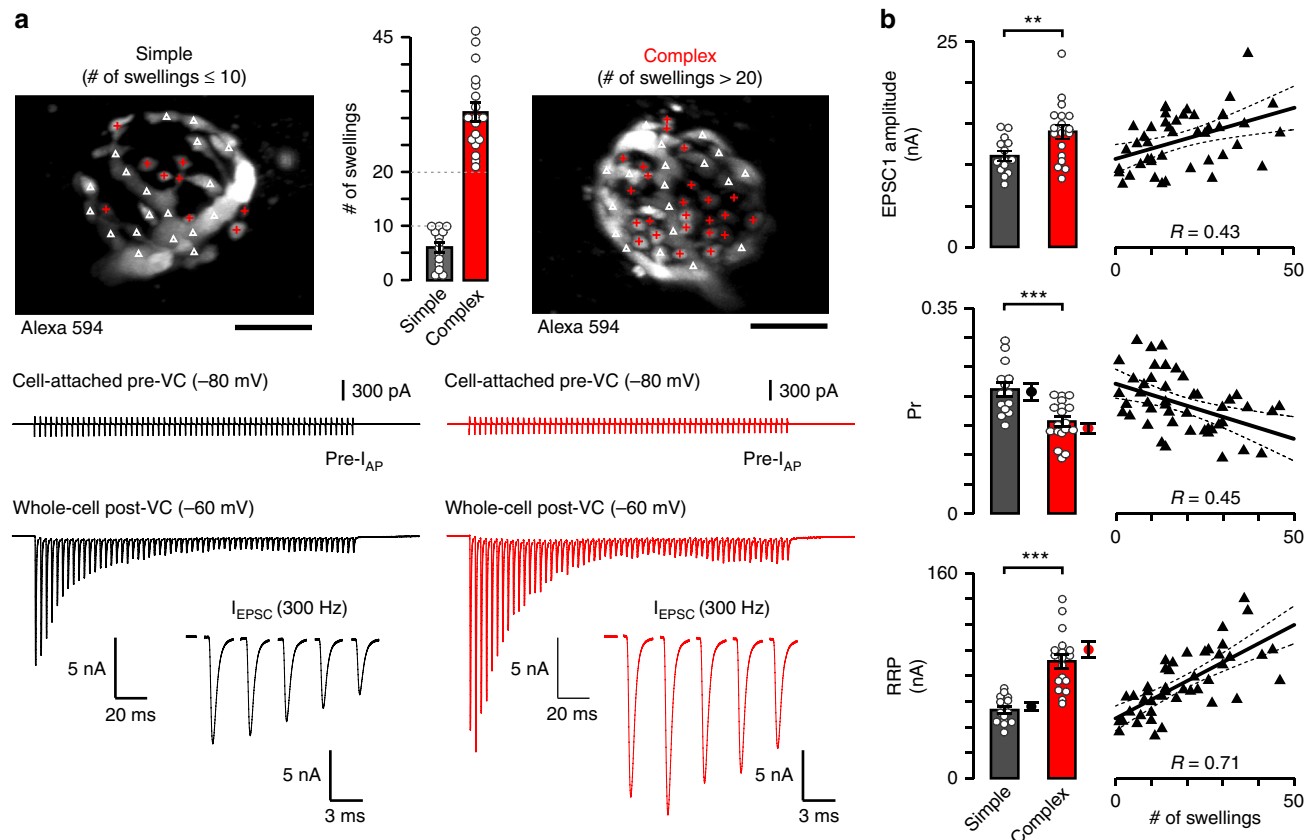

**Fig. 1** Mature calyx of Held synapses with diverse morphologies exhibit distinct synaptic function. **a** Simple (≤10 swellings, black) and complex (>20 swellings, red) calyces are visualized in z-axis projection images (Alexa 594, 60 μM; stalks, white arrowheads; swellings, red crosses). Bar graphs show the average number of swellings (simple, 6.1 ± 0.91, $n = 14/11$, complex, 31.1 ± 1.73, $n = 18/16$, $t = 12.762$, $p < 0.0001$, df = 25, unpaired $t$-test with Welch correction [u.t.t.W.]). Representative recordings (presynaptic cell-attached and postsynaptic whole-cell voltage-clamp [VC]) demonstrate the action potential train-evoked EPSCs (300 Hz, 200 ms, extracellular axonal stimulation). Insets: EPSC 1–5. Stimulation artifacts were removed. **b** Bar graphs outline the synaptic properties of simple and complex calyces (EPSC1 amplitude, simple, 11.11 ± 0.58 nA, $n = 14/11$, complex, 14.10 ± 0.86 nA, $n = 18/16$, $t = 2.872$, $p = 0.008$, df = 28, u.t.t.W.; Pr, release probability, simple, 0.212 ± 0.012, $n = 14/11$, complex, 0.157 ± 0.008, $n = 18/16$, $t = 3.967$, $p = 0.0004$, df = 30, unpaired $t$-test; RRP, size of readily releasable pool, simple, 53.5 ± 2.84 nA, $n = 14/11$, complex, 91.8 ± 5.31 nA, $n = 18/16$, $t = 6.351$, $p < 0.0001$, df = 25, u.t.t.W.). Dot-plots (right; Sw, swelling) of individual experiments were fitted with linear function (mean ± 95% confidence interval) to show the trends. Filled circles next to the bar graphs represent mean ± SEM of Pr and RRP estimated with non-linear synaptic replenishment considered (Pr, simple, 0.207 ± 0.014, $n = 14/11$, complex, 0.145 ± 0.008, $n = 18/16$; $t = 3.87$, $p = 0.0009$, df = 21, u.t.t.W.; RRP, simple, 55.6 ± 3.47 nA, $n = 14/11$, complex, 99.9 ± 5.98 nA, $n = 18/16$; $t = 6.402$, $p < 0.0001$, df = 26, u.t.t.W.). $R$ demonstrates correlation strength. Bar graphs summarize mean ± SEM (**$p < 0.01$, ***$p < 0.001$). See Supplementary Figure 1 and 2. Scale bars: 10 μm (**a**)

$I_{AP}$ from a stalk ($n = 8$ stalks/6 mice), we always recorded pre-$I_{AP}$ from swelling(s) of the same calyx ($n = 11$ swellings/6 mice; in one of the calyces we managed to record 1 stalk and 2 swellings, in another calyx 1 stalk and 3 swellings), indicating that APs do not fail to propagate into different compartments.

Due to the lack of $Na^+$ channels outside the heminode, APs propagate passively into the calyx and may undergo changes in their waveform[39,40]. Since the time interval between the inward and outward deflection of pre-$I_{AP}$ (Fig. 2a; peak-to-peak time) approximates the half-width of APs[38,41], we used pre-$I_{AP}$ to compare AP shape in different compartments. The average peak-to-peak time was independent of the location (Fig. 2a, b). The lag of peak from stalk to swelling or from stalk to stalk was not different (Fig. 2c). These data suggest that APs propagate throughout the calyx without changes in the width and timing, ensuring the synchronized activation of VGCCs in swellings and stalks.

As many swellings are too small to be patch-clamped, we further confirmed by $Ca^{2+}$ imaging that APs reliably propagate to

evoke $Ca^{2+}$ influx in stalks and swellings. We loaded the terminals with Fluo-4 (50 μM, Green, $K_D = 345$ nM) and Alexa 594 (15 μM, Red, morphological tracer) through a patch electrode, and performed two-photon laser scanning microscopy (TPLSM). APs were elicited by current injection in whole-cell configuration or afferent stimulation using bipolar electrode. $Ca^{2+}$ transients were expressed as the ratio of two colors ($\Delta G/R$; see Methods). By imaging at least one swelling and a neighboring stalk simultaneously (Supplementary Figure 3a), we found that the $Ca^{2+}$ transients were detectable throughout the calyces, equal in decay time ($\tau_{decay}$) but larger in amplitude and integral in swellings than stalks (amplitude×$\tau_{decay}$, a measure independent of buffer capacity[42], Supplementary Figure 3b). This difference in the magnitude of $Ca^{2+}$ transients in the two compartments may originate from differences in the surface to volume ratio and/or the number of VGCCs. These data indicate that single APs reliably propagate into swellings and stalks to evoke $Ca^{2+}$ influx, and that signaling downstream from APs likely accounts for synaptic heterogeneity.

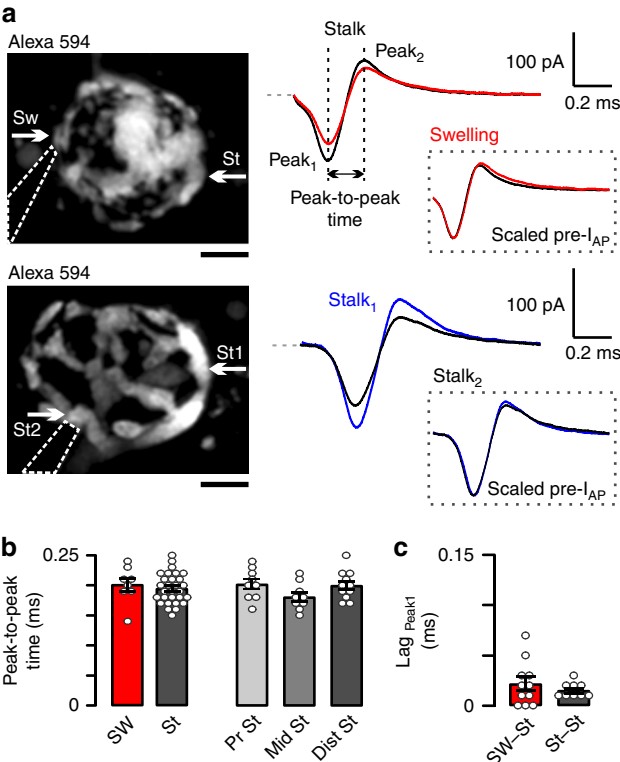

**Fig. 2** Action potential waveforms are similar throughout the calyx. **a** Calyces are visualized in z-axis projection images (Alexa 594, 60 μM; dashed lines: cell-attached pipette). White arrows indicate the sites of presynaptic compound action potential current (pre-$I_{AP}$) recordings (Sw swelling, St stalk). Stimulation artifacts were removed (dashed lines show baselines). Peak-to-peak time is the interval between peak$_1$ and peak$_2$ of pre-$I_{AP}$ (vertical lines and black arrow). Inset: amplitude scaled pre-$I_{AP}$-s in swellings and stalks. **b–c** Bar graphs summarize peak-to-peak times in swellings and stalks (Sw, 0.201 ± 0.011 ms, $n = 8/7$, St, 0.195 ± 0.005 ms, $n = 29/15$, $t = 0.612$, $p = 0.545$, df = 35, unpaired $t$-test) or at different distances from heminode in stalk (Pr St, proximal stalk, 0.202 ± 0.009 ms, $n = 9/9$, Mid St middle stalk, 0.181 ± 0.008 ms, $n = 9/5$, Dist St distal stalk, 0.200 ± 0.007 ms, $n = 11/7$, $p = 0.141$, $F_{2,26} = 2.12$, one-way ordinary ANOVA). Lags are calculated as delay between peak$_1$-s at different locations (stalk to swelling, 0.0218 ± 0.0067 ms, $n = 11/7$, stalk to stalk, 0.0156 ± 0.0024 ms, $n = 9/8$; $t = 0.877$, $p = 0.398$, df = 12, unpaired $t$-test with Welch correction). Bar graphs summarize mean ± SEM. Scale bars: 5 μm (**a**)

## $Ca^{2+}$ current density scales with morphological complexity.

Because Pr and RRP size are dependent on AP-driven $Ca^{2+}$ influx into the terminal[17,23,43,44], we measured the $Ca^{2+}$ currents ($I_{Ca2+}$) in simple and complex calyces (see Methods). We delivered 20 ms long voltage steps from a −100 mV baseline in 10 mV increments from −90 to +50 mV. After each recording we imaged the calyx morphology using TPLSM in situ (Fig. 3a). As the I-V curves of step and tail $I_{Ca2+}$-s of simple and complex calyces perfectly overlapped (see the scaled trace, Fig. 3a and Supplementary Figure 4a), we concluded that the voltage-dependence of VGCCs is identical in the two groups. The step $I_{Ca2+}$, typically activated at −40 mV, had a maximum at −10 mV and reversed at +50 mV in both calyx subgroups (Fig. 3a). The tail $I_{Ca2+}$ activated at −40 mV and reached a maximum at +10 to +20 mV indicating a maximal relative open probability above this voltage (Supplementary Figure 4a). We found that the maximal $I_{Ca2+}$ amplitudes were significantly larger in complex than simple calyces and correlated strongly with the number of

swellings (Fig. 3a, b and Supplementary Figure 4a,b). The whole-cell capacitance increased in parallel with the calyx complexity ($C_{whole-cell}$; Fig. 3c) and was used to calculate $I_{Ca2+}$ densities of individual calyces (Fig. 3b and Supplementary Figure 4b). We found that the positive correlation between the $I_{Ca2+}$ density and the number of swellings remains robust, consistent with the idea that the larger number of release sites in complex calyces are due to the presence of larger number of swellings that are functional modules for transmitter release.

## Diverse VGCC cluster topography in morphological modules.

Since SDS-FRL of the calyx of Held terminals showed that VGCCs are locally organized into clusters containing variable numbers of VGCCs with different Pr and coupling distances[27], we considered the possibility that stalks and swellings may have different topographical arrangements of VGCCs. We took advantage of a knock-in mouse line in which the N-terminus of P/Q-type VGCC $\alpha_1$ subunit, the predominant subtype in mature calyx[45], was tagged with citrine, a GFP variant ($Cacna1a^{Citrine}$, [46]). First, we traced the presynaptic terminal by Alexa 594 dextran to identify the morphological modules then fixed the tissue for confocal imaging (Fig. 4a). After acquiring and de-convolving z-stack images, we observed distinct fluorescence puncta in swellings and stalks (Fig. 4a–c; see profile line and plot in Fig. 4b) and on the release face of calyces (Fig. 4c) consistent with previous electron microscopy data about VGCC cluster localization[27]. We found higher number of puncta in complex calyces (Fig. 4d; 509 ± 31, simple calyx, 709 ± 27, complex calyx; mean ± standard error of mean). These quantities are in line with the number of AZs on calyces determined by electron microscopy[27,33]; (see the cluster separation in Fig. 4b). There is a strong and positive correlation between the number of puncta and swellings (Fig. 4d), yielding 432 clusters on stalks (y-axis intercept) and 10.4 clusters per swellings (slope). These results suggest that the swelling-rich complex calyces contain additional release sites to support high-fidelity neurotransmission[4].

To explore the distribution of clusters and their size (number of VGCCs per cluster) within the two morphological modules, we paired the swellings and stalks in equal depth from the slice surface with a clear top view to avoid variation in excitation power and emission detection efficiency due to tissue scattering. We quantified the number of clusters per contact area and found that the cluster number within swellings is higher than in stalks (Fig. 4e; swelling, 2.1 ± 0.07 μm$^{-2}$; stalk, 1.5 ± 0.06 μm$^{-2}$). To compare the number of VGCCs per cluster within the paired morphological modules we took advantage of the 100% labeling efficiency of the knock-in mice and quantified the cumulative fluorescence intensities per cluster in swelling and stalk (or the sum of fluorescence intensities of all voxels in the cluster). Since the citrine fluorescence intensities were uncalibrated, we expressed the difference between the morphological modules as relative values (stalk/swelling, see Methods). We found ~35% larger cumulative intensity per cluster in stalks inferring more VGCCs per clusters (Fig. 4f). As predicted by the perimeter release model[27], fewer VGCCs per cluster in swellings would yield a smaller local $[Ca^{2+}]$ and lower Pr than in stalks. As a consequence of increasing the number of swellings with lower Pr, the global (whole-terminal) Pr decreases.

Because the size of VGCC clusters was larger within stalks than swellings, we assumed that the whole-terminal functional diversity is continuously tuned by varying the proportion of low and high Pr clusters in complex and simple calyces, as indicated by the positive correlation between swellings and morphological complexity. We considered a model where the number of high Pr stalk clusters is constant and only the number

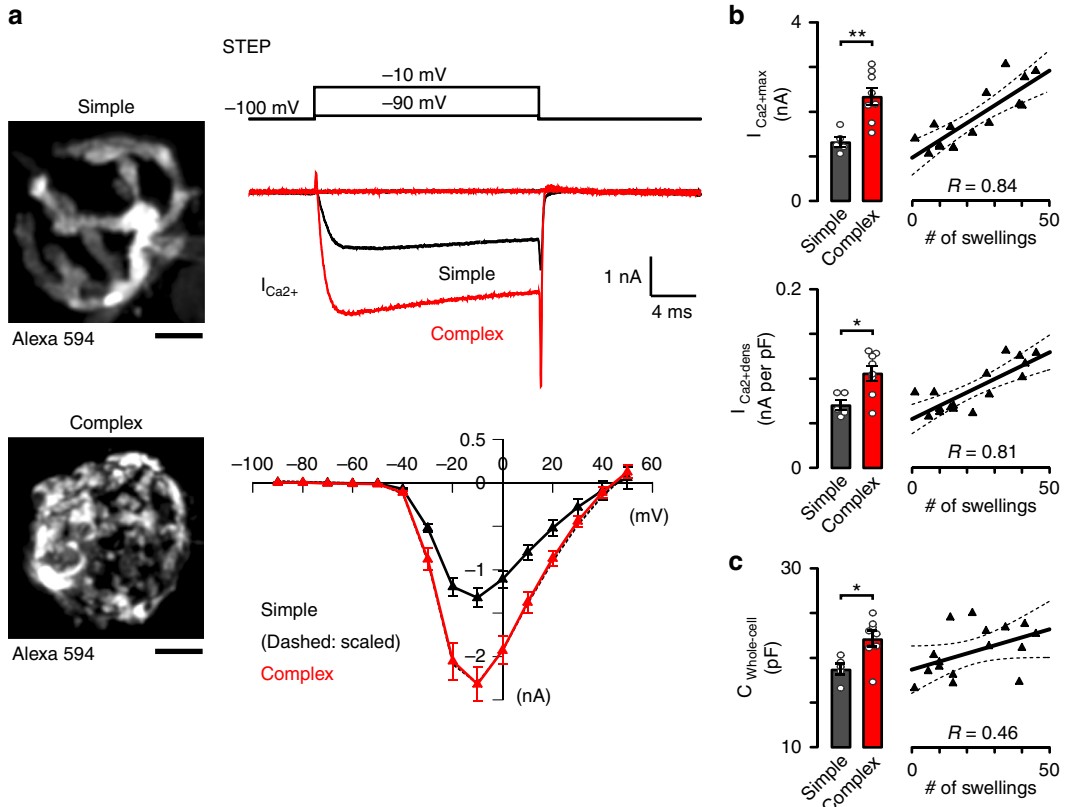

**Fig. 3** $Ca^{2+}$ current density is larger in complex calyces. **a** Example calyces are visualized in *z*-axis projection images. Representative traces show presynaptic $Ca^{2+}$ currents ($I_{Ca2+}$) recorded from simple (black) and complex (red) calyces kept at −100 mV. $I_{Ca2+}$-s were tested by voltage steps between −90 and +50 mV with 10 mV increments (see schematic drawing). Average voltage-dependence curves of step $I_{Ca2+}$-s (mean ± SEM) are shown in simple and complex calyces. Amplitude-scaled version of simple calyx I–V curve (dashed black) overlaps with the complex one. **b**, **c** Bar graphs show the maximal step $I_{Ca2+}$ amplitude ($I_{Ca2+max}$), step $I_{Ca2+}$ density ($I_{Ca2+dens}$), and whole-cell capacitance ($C_{whole-cell}$) of simple and complex calyces (step $I_{Ca2+}$ amplitude, simple, 1.33 ± 0.11 nA, $n = 5/4$, complex, 2.34 ± 0.20 nA, $n = 8/7$, $t = 3.846$, $p = 0.003$, df = 11, unpaired *t*-test [u.t.t]; step $I_{Ca2+}$ density, simple, 0.071 ± 0.0057 nA per pF, $n = 5/4$, complex, 0.106 ± 0.0087 nA per pF, $n = 8/7$, $t = 2.974$, $p = 0.013$, df = 11, u.t.t.; $C_{whole-cell}$, simple, 18.8 ± 0.62 pF, $n = 5/4$, complex, 22.2 ± 0.83 pF, $n = 8/7$, $t = 2.918$, $p = 0.014$., df = 11, u.t.t.). Scatter-plots of individual experiments (right) were fitted with linear function (mean ± 95% confidence interval). *R* demonstrates correlation strength. Bar graphs summarize mean ± SEM (*$p < 0.05$, **$p < 0.01$). See Supplementary Figure 4. Scale bars: 5 μm (**a**)

of low Pr swelling clusters varies because the RRP systematically increased with the number of swellings. To estimate Pr in stalk ($Pr_{St}$) and swellings ($Pr_{Sw}$) we fitted the whole-calyx Pr scatter-plot ($Pr_{WholeCalyx}$; Figs. 1b, 4g) with the weighted mean of Pr-s of distinct AZs on the entire stalk ($n_{AZSt}$) and per swelling ($n_{AZSw}$),

$$\frac{Pr_{St} \times n_{AZSt} + Pr_{Sw} \times n_{AZSw} \times n_{Sw}}{n_{AZSt} + n_{AZSw} \times n_{Sw}}, \qquad (1)$$

where $n_{Sw}$ is the number of swellings. We approximated $n_{AZSt}$ and $n_{AZSw}$ with the number of VGCC clusters on entire stalk and per swelling (see the linear fit in Fig. 4d) because the numbers of clusters in simple and complex calyces are in line with the reported range of the number of AZs per calyx[33,47] and are consistent with the observation that one cluster localizes to one AZ[27,33]. We estimated that $Pr_{St}$ and $Pr_{Sw}$ are 0.233 ± 0.031 and 0.069 ± 0.071 (mean ± 95% confidence interval [CI]), respectively.

The EPSC1 amplitude (Fig. 4h) was the product of Pr and RRP on stalk and swelling,

$$Pr_{St} \times RRP_{St} + Pr_{Sw} \times RRP_{Sw} \times n_{Sw}, \qquad (2)$$

where $RRP_{St}$ and $RRP_{Sw}$ are the RRP sizes in entire stalk and per swelling, respectively, and were approximated by linear regression of the whole-calyx RRP scatter-plot (Fig. 1b; *y*-intercept, $RRP_{St}$, 46.7 ± 9.69 nA, mean ± 95% CI; slope, $RRP_{Sw}$, 1.46 ± 0.44 nA,

mean ± 95% CI). Our calculation for synaptic strength (see simulated EPSC1 amplitude in Fig. 4g), 11.46 nA for simple calyces with 6 swellings and 13.97 nA for complex calyces with 31 swellings, was comparable to the real EPSC amplitudes with the same average number of swellings (Fig. 1, see linear fit in Fig. 4h). This formula, which includes a relative number of low and high Pr clusters proportional to the number of swellings, predicts well the correlation between synaptic strength and morphological complexity.

**VGCC-SV coupling distances differ in swellings and stalks.** Because the VGCC-SV coupling distance critically determines Pr[17,23], we hypothesized that the coupling distance is longer in complex than simple calyces. To test this we examined the fractional block of EPSCs by EGTA, a $Ca^{2+}$ chelator with slow on-rate for $Ca^{2+}$ binding[48]. We performed presynaptic cell-attached recordings with 10 mM EGTA in the pipette and evoked EPSCs by axonal stimulation (control). We subsequently broke through the presynaptic patch membrane, allowing EGTA to diffuse into the terminal for 3–5 min. Then we slowly withdrew the presynaptic pipette to reseal the terminal. With this method we could preserve synaptic transmission (e.g. glutamate[34]) while ensuring an efficient chelator delivery[24]. We found that the EGTA effect on attenuating EPSC amplitude is strongly correlated with the

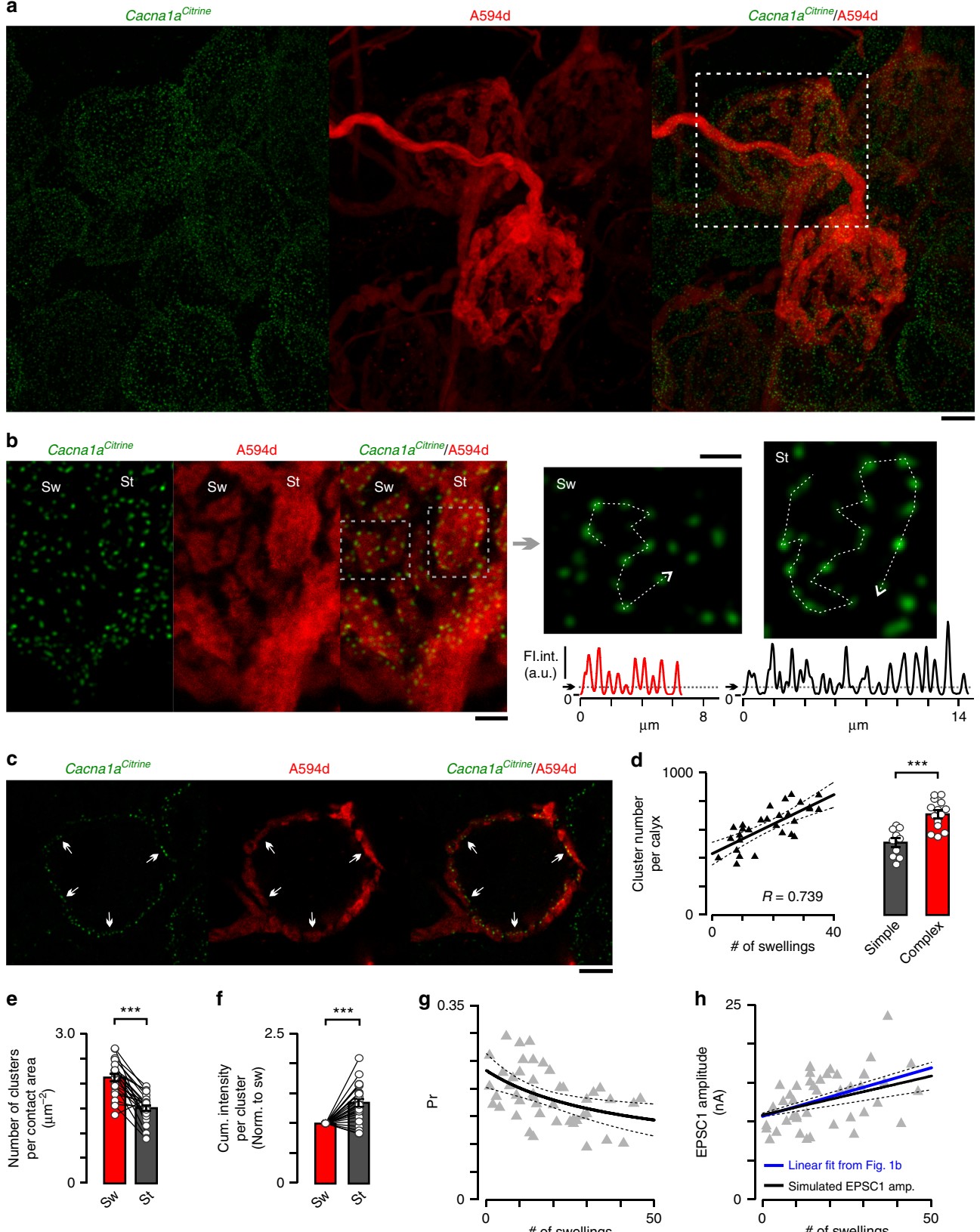

number of swellings, being significantly larger in complex (38.4%) than simple calyces (19.8%; Fig. 5a–c). This result suggested that the coupling distance is longer in swellings than stalks.

VGCC-SV coupling distances can influence the synaptic delay (SD;[27]). We compared the delays in simple and complex calyces (see SD$_1$ and SD$_2$ in Fig. 5a) and found no detectable difference (Supplementary Figure 5a and Supplementary Table 1). We attribute this to the omnipresent high Pr release sites which dominate the first latency to release and thus preferentially influence the delay. Previously, EPSC time course has been found

**Fig. 4** Topography of VGCC clusters is different in swellings and stalks. **a** Z-axis projection images visualize citrine-tagged Cav2.1 $\alpha_1$ subunits (*Cacna1a*$^{Citrine}$, green) and calyces (Alexa594 dextran, A594d; red). Calyx highlighted with white dashed rectangle is shown in **b** and **d**. **b** Calyx top view (z-axis projection, 7 optical sections) shows punctate citrine labeling (Sw, swelling, St, stalk). Areas in gray dashed rectangles are enlarged on the right. Profile line and plot demonstrate separation of puncta (Sw, red; St, black). Thresholds (black arrows) identify clusters. **c** Calyx cross-section shows clusters along the presynaptic face (white arrows). **d** The number of clusters per calyx versus calyx complexity plot was fitted linearly (y-axis intercept: number of clusters on stalk, 431.9 ± 79.7; slope: number of clusters per swelling, 10.36 ± 3.8; mean ± 95% confidence interval [CI]). Bar graphs summarize the cluster number per calyx (simple, 509.4 ± 31.1, $n = 10/7$, complex, 708.8 ± 27.0, $n = 14/4$, $t = 4.821$, $p < 0.0001$, df = 22; unpaired t-test). **e** The cluster number per contact area was quantified in swellings and stalks (Sw, 2.12 ± 0.070 μm$^{-2}$, $n = 25/4$, St, 1.51 ± 0.057 μm$^{-2}$, $n = 25/4$, $t = 7.680$, $p < 0.0001$, df = 24; paired t-test). **f** Cumulative fluorescence intensity per cluster (correlate of VGCC number per cluster) is expressed as relative value (St/Sw, 1.347 ± 0.063, $n = 25/4$, $t = 5.488$, $p < 0.0001$, df = 24, one sample t-test). **g**, **h** Individual release probabilities (Pr) and EPSC1 amplitudes for all recorded synapses were dot-plotted (gray triangles). Note the identical scatter-plots in Fig. 1b. **g** Whole-calyx Pr plot was fitted with the weighted mean of Pr-s of distinct active zones (black, see Equation (1); mean ± 95% CI). **h** Simulated synaptic strength (black, see Equation (2); mean ± 95% prediction band) closely predicts real EPSC amplitudes (blue, linear fit). Bar graphs summarize mean ± SEM (***$p < 0.001$). R demonstrates correlation strength. Scale bars: 5 μm (**a**), 2 μm (**b**, left), 1 μm (**b**, right), 5 μm (**c**)

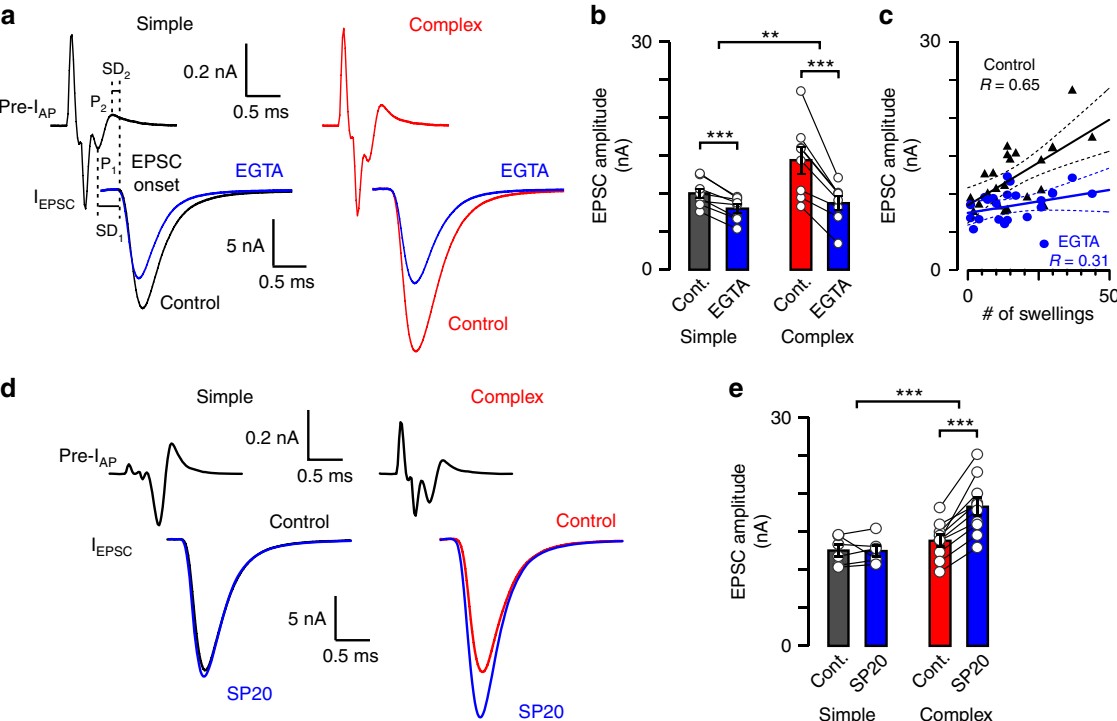

**Fig. 5** Synaptic strength is more sensitivity to EGTA and downregulated by septin 5 in complex calyces. **a** Presynaptic compound action potential currents (pre-I$_{AP}$) and EPSCs (I$_{EPSC}$; axonal stimulation) were tested in control (black, simple calyx; red, complex calyx) and after EGTA injection (blue, 10 mM, 3–5 min loading). SD$_1$ (interval between peak$_1$ of pre-I$_{AP}$ and EPSC onset) and SD$_2$ (interval between peak$_2$ of pre-I$_{AP}$ and EPSC onset; dashed lines) are synaptic delays. **b** Bar graphs show EGTA effect on EPSC amplitudes (simple, control, 10.05 ± 0.64 nA, EGTA, 8.06 ± 0.55 nA, $n = 8/7$, $t = 5.762$, $p = 0.0007$, df = 7, paired t-test [p.t.t.]; complex, control, 14.44 ± 1.76 nA, EGTA, 8.77 ± 0.93 nA, $n = 8/8$, $t = 5.558$, $p = 0.0009$, df = 7, p.t.t.). The difference in EGTA effect on simple and complex calyces was quantified as EGTA/control (simple, 0.802 ± 0.032, $n = 8/7$, complex, 0.616 ± 0.042, $n = 8/8$, $t = 3.552$, $p = 0.0032$, df = 14, unpaired t-test [u.t.t.]). **c** EPSC amplitudes for all EGTA injected synapses were plotted against swelling number (black triangles, control; blue dots, EGTA) and fitted linearly (mean ± 95% confidence interval). **d** Pre-I$_{AP}$ and I$_{EPSC}$ were tested in control (black, simple calyx; red, complex calyx) and after septin 5 antibody injection (blue, SP20, 1:1000, 3–5 min loading). **e** Bar graphs show SP20 effect on EPSC amplitudes 10–15 min after presynaptic pipette removal (simple, control, 12.53 ± 0.77 nA, SP20, 12.46 ± 0.70 nA, $n = 6/4$, $t = 0.1137$, $p = 0.914$, df = 5, p.t.t.; complex, control, 13.83 ± 0.76 nA, SP20, 18.29 ± 1.17 nA, $n = 10/8$, $t = 6.808$, $p < 0.0001$, df = 9, p.t.t.). The difference in SP20 effect on simple and complex calyces was quantified as SP20/control (simple, 1.002 ± 0.045, $n = 6/4$, complex, 1.323 ± 0.043, $n = 10/8$, $t = 4.915$, $p = 0.0002$, df = 14, u.t.t.). R demonstrates correlation strength. Bar graphs summarize mean ± SEM (**$p < 0.01$, ***$p < 0.001$). See Supplementary Figure 5

stable even when the release rate was reduced[17,23] suggesting that quantal events during an AP are highly synchronized across AZs. Our data support this because (a), the miniature and evoked EPSC time courses were comparable; (b), the EPSC time courses of simple and complex calyces were not different; and (c), EGTA injection did not affect the EPSC time course differentially in simple and complex calyces (Supplementary Figure 2, Supplementary Figure 5b and Supplementary Table 1).

**Septin 5 diversifies the VGCC-SV coupling distance**. We previously demonstrated that the filamentous protein septin 5 (Sept5) acts as a physical barrier to keep SVs more distant from the AZs in the immature calyx, and relocates during development[49,50]. We therefore considered the possibility that Sept5 may differentiate the two release topographies with discrete coupling distances. Sept5 antibody was injected into the terminals (SP20, 1:1000, 3–5 min). SP20 is known to disrupt the structure

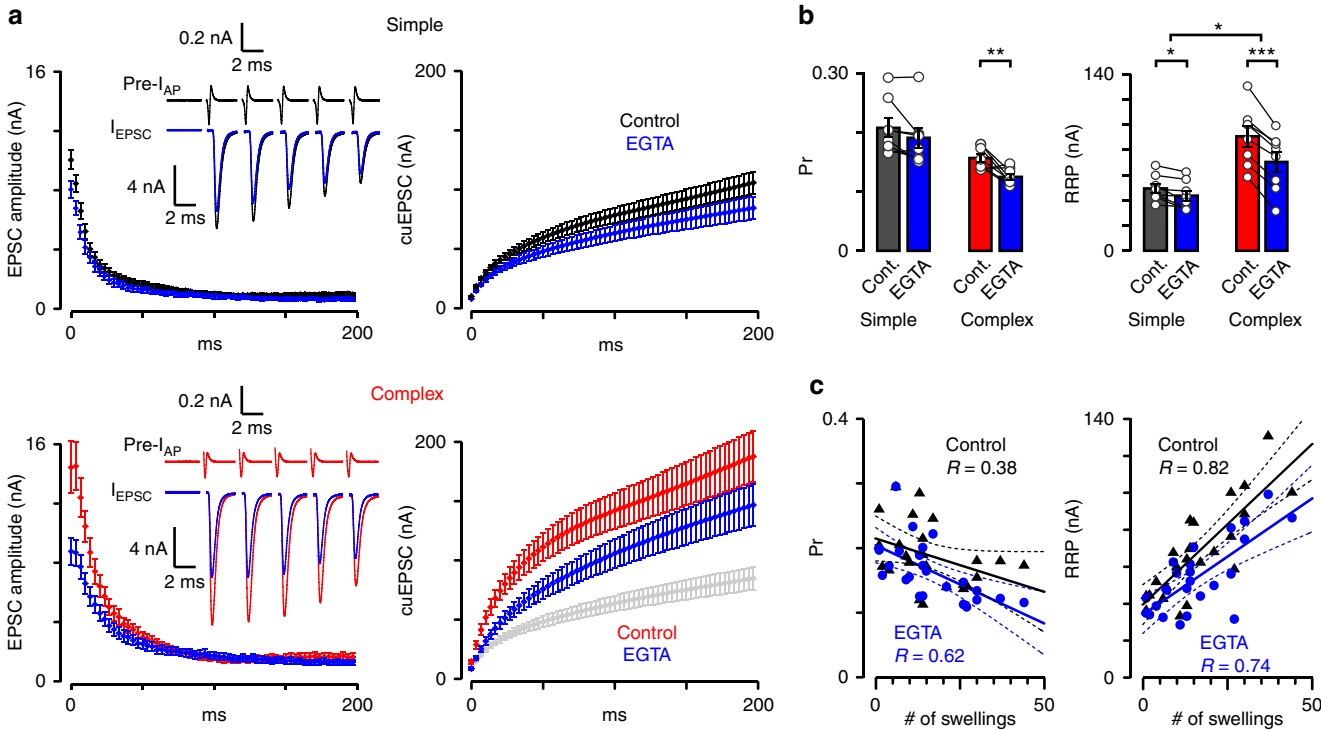

**Fig. 6** EGTA sensitivity of the Pr and size of RRP is heterogeneous in mature calyces. **a** Average traces (mean ± SEM) show amplitudes (left panels) and cumulative amplitudes (cuEPSC, right panels) of action potential (AP) train-evoked EPSCs (300 Hz, 200 ms) in simple (black, upper) and complex (red, lower) calyces before and after EGTA loading (blue, 10 mM, 3–5 min). Insets: representative pre-$I_{AP}$ 1–5 (presynaptic compound AP currents) and EPSC1–5. Gray trace (last panel): cuEPSC of simple calyces after EGTA injection for comparison (mean ± SEM). **b** Bar graphs show release probability (Pr) and readily releasable pool (RRP) before and after EGTA loading. (Pr, simple, control, 0.208 ± 0.016, EGTA, 0.191 ± 0.017, $n = 8/7$, $t = 2.211$, $p = 0.0627$, df = 7, paired $t$-test [p.t.t.]; complex, control, 0.157 ± 0.007, EGTA, 0.125 ± 0.005, $n = 8/8$, $t = 3.53$, $p = 0.0096$, df = 7, p.t.t.; RRP, simple, control, 49.32 ± 3.56 nA, EGTA, 43.56 ± 3.904 nA, $n = 8/7$, $t = 3.412$, $p = 0.0113$, df = 7, p.t.t.; complex, control, 90.64 ± 8.13 nA, EGTA, 70.32 ± 7.97 nA, $n = 8/8$, $t = 9.363$, $p < 0.0001$, df = 7, p.t.t.). The difference in EGTA effect on simple and complex calyces was quantified as EGTA/control (Pr, simple, 0.917 ± 0.035, $n = 8/7$, complex, 0.813 ± 0.052, $n = 8/8$, $t = 1.664$, $p = 0.118$, df = 14, unpaired $t$-test [u.t.t.]; RRP, simple, 0.881 ± 0.037, $n = 8/7$, complex, 0.762 ± 0.036, $n = 8/8$, $t = 2.305$, $p = 0.037$, df = 14, u.t.t.). **c** Pr and RRP size for all EGTA loaded synapses were plotted against the number of swellings (black triangles, control; blue dots, EGTA) and linearly fitted (mean ± 95% confidence interval). The $y$-intercept of the RRP fit gives RRP size in stalk (39.30 ± 10.08 nA), the slope determines RRP per swelling (1.74 ± 0.55 nA). $R$ demonstrates correlation strength. Bar graphs summarize mean ± SEM (*$p < 0.05$, **$p < 0.01$, ***$p < 0.001$). See Supplementary Figure 6

and function of septin filaments efficiently[50]. We found that SP20 potentiated the synaptic strength only in complex calyces (Fig. 5d, e), implicating Sept5 as a molecular spacer underlying longer coupling distances in swellings than stalks, thereby linking structural and functional heterogeneity.

**Diversity in VGCC-SV spatial coupling tunes Pr, RRP, and STP.** To investigate how the diversity in VGCC-SV topography affects Pr, RRP, and STP of calyces with different morphologies, we injected EGTA into the terminal to reduce the extent of $Ca^{2+}$ domains and attenuate the release of SVs with longer coupling distance. We recorded EPSCs by stimulating the axon fiber with a pulse train (300 Hz, 200 ms), and constructed the cumulative EPSC amplitude to determine Pr and RRP size (Fig. 6a). In simple and complex calyces, EGTA reduced the Pr by 8.3% and 18.7%, and the size of RRP by 11.9% and 23.8%, respectively (Fig. 6a, b). Moreover, the correlations between the quantal parameters and the number of swellings for all EGTA injected synapses (Fig. 6c), and their heterogeneous EGTA sensitivity implicate two subpools of SVs in RRP with different Pr, one coupled tighter, the other coupled more loosely to the VGCCs[23,51,52], and differentially distributed in swellings and stalks.

Elevation of residual $Ca^{2+}$ level during neural activity is a well-established mechanism of STP[8,53] thus we hypothesized that it might be responsible for the different pattern of STP: simple

calyces with higher Pr purely depress, complex calyces with lower Pr first facilitate then depress (Fig. 1 and Supplementary Figure 1). To test this, we quantified PPR in simple and complex calyces before and after EGTA injection (10 mM, 3–5 min), however, EGTA did not affect the PPR (Fig. 6a and Supplementary Figure 6a). This is likely because, (a) STF in calyx is partly elicited by EGTA-insensitive $I_{Ca2+}$ facilitation;[54] (b) the decrease in initial Pr results in a tendency for facilitation, while the attenuation of residual $Ca^{2+}$ buildup counteracts facilitation. Next, we approximated the EGTA effect on SV depletion, the main mechanism of STD in mature calyx[31–36]. EGTA significantly slowed the STD affecting most the slowly releasing subpool of SVs in the complex calyces (Supplementary Figure 6b,c and Supplementary Table 2). When we compared the sizes of subpools, we found that only the slowly releasing subpool decreased significantly after EGTA application and preferentially in complex calyces, indicating that these SVs, more frequent in complex calyces (i.e. swellings), are placed more distal from the VGCCs than the SVs in the fast-releasing subpool (i.e. stalks; Supplementary Figure 6b,c and Supplementary Table 2;[55,56]).

**Morphological diversity tunes postsynaptic spiking fidelity.** Because we found larger size of RRP and longer VGCC-SV coupling distance in complex calyces, we examined their impact on the functional output of the synapse during sustained high

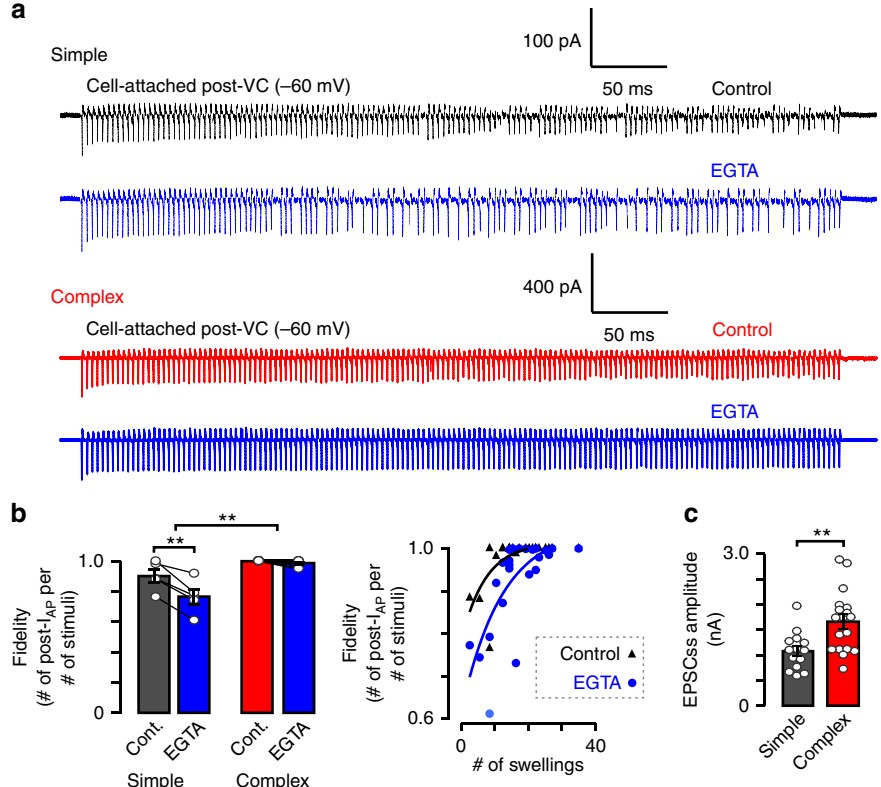

**Fig. 7** Postsynaptic spiking is less reliable and more sensitive to EGTA in simple calyces. **a** Representative pre- and postsynaptic cell-attached recordings of simple (black) and complex (red) calyces show spiking in response to axonal stimulation (300 Hz, 500 ms) before (control) and after presynaptic EGTA loading (blue; 10 mM, 3–5 min). **b** Bar graphs summarize fidelity (number of successful postsynaptic compound action potential currents [post-$I_{AP}$] per number of stimuli) in control and after EGTA loading (simple, control, 0.902 ± 0.042, EGTA, 0.767 ± 0.049, $n = 5/5$, $t = 5.612$, $p = 0.005$, df = 4, paired $t$-test [p.t.t.]; complex, control, 1.00 ± 0.00, EGTA, 0.988 ± 0.006, $n = 8/8$, $t = 1.891$, $p = 0.100$, df = 7, p.t.t.). The difference in EGTA effect on simple and complex calyces was quantified as EGTA/control (simple, 0.849 ± 0.027, $n = 5/5$, complex, 0.988 ± 0.006, $n = 8/8$, $t = 5.123$, $p = 0.007$, df = 4, unpaired $t$-test with Welch correction [u.t.t.W.]). Individual fidelity values were plotted against the number of swellings (black triangles, control; blue dots, EGTA) and fitted with single exponential. **c** Bar graphs summarize mean ± SEM (**$p < 0.01$) of control steady-state EPSC amplitudes (EPSC$_{ss}$, simple, 1.08 ± 0.10 nA, $n = 14/11$, complex, 1.66 ± 0.14 nA, $n = 18/16$, $t = 3.337$, $p = 0.0024$, df = 28, u.t.t.W.). See Supplementary Figure 7

frequency transmission (300 Hz, 500 ms) before and after loading the calyx with EGTA (10 mM, 3–5 min). We found that simple calyces drive spikes with 90.2% fidelity (Fig. 7a, b) with the failures first appearing in the 100–200 ms segment of the train (Supplementary Figure 7) while complex calyces did not fail at all (Fig. 7a, b) indicating that larger RRP endows complex calyces with larger fidelity. Contrary to the prediction that the spiking fidelity of complex calyces is more sensitive to EGTA, complex calyces were unperturbed, whereas EGTA shortened the delay for the onset of the first failure and decreased the fidelity of postsynaptic spiking in synapses driven by simple calyces (Fig. 7a, b and Supplementary Figure 7). Because a small change in the postsynaptic depolarization around the AP generation threshold can have a dramatic effect on spiking, we quantified the steady-state EPSC (EPSC$_{ss}$) amplitude and found that the absolute EPSC$_{ss}$ amplitude in simple calyces is significantly smaller than that in complex calyces (Fig. 7c), in line with the absolute difference in the RRP between simple and complex calyces after EGTA injection (Fig. 6a, b; see gray trace in last panel of Fig. 6a). We suggest that simple calyces with tighter coupling can drive the early spiking but fail to sustain firing as a result of their limited number of SVs in the slowly releasing subpool (and blocked Ca$^{2+}$-dependent replenishment in EGTA). Complex calyces with longer coupling distance and larger subpool of slowly releasing SVs in swellings maintain the synaptic drive required to depolarize the principal neurons close to the spike generation

threshold, despite partial block by EGTA. We conclude that increasing morphological complexity expands the size of RRP by increasing the number of AZs and the coupling distance in the terminal to sustain transmitter release during high-frequency neurotransmission. By varying the morphological complexity, synapses tune to various levels of fidelity, endowing a single synapse population with diverse filtering properties.

**Ca$^{2+}$ transients are diverse in the morphological modules.** Given the significant differences in RRP size and EGTA-sensitivity between simple and complex calyces (in addition to the larger number of AZs), we postulated that the larger Ca$^{2+}$ accumulation in swellings than stalks during high-frequency spiking may warrant the reliability of transmitter release in complex calyces because SVs at variable distances from VGCCs can be recruited for release[23,31,43,52,57]. To test the kinetics and magnitude of Ca$^{2+}$ accumulation in the morphological modules we loaded the calyces with a low affinity Ca$^{2+}$ indicator (Fluo-4FF, 50 μM, $K_D = 9.2$ μM, Green) and Alexa 594 (15 μM, Red; Fig. 8) by preloading the patch electrode. AP trains (100–300 Hz, 200 ms) were elicited by current injection in current-clamp mode or afferent stimulation using bipolar electrode. Ca$^{2+}$ transients were expressed as the ratio of two colors ($\Delta G/R$; see Methods). We found larger and faster Ca$^{2+}$ transients in swellings than stalks (Fig. 8a, b and Supplementary Table 3). As the magnitude

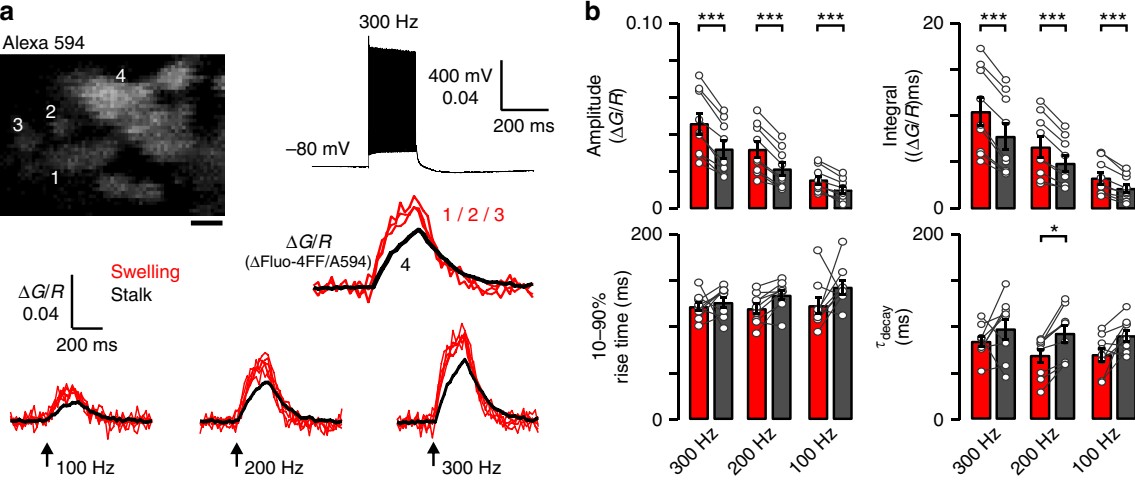

**Fig. 8** Action potential train-evoked $Ca^{2+}$ transients are larger and faster in swellings than stalks. **a** Representative image outlines a calyx loaded with Fluo-4FF (50 μM; $K_D = 9.2$ μM) and Alexa 594 (15 μM) and frame-scanned by a two-photon laser scanning microscope. Action potential (AP) trains (100–300 Hz, 200 ms) were elicited by current injection in current-clamp mode or afferent stimulation using bipolar electrode. $Ca^{2+}$ transients were recorded in swellings (location 1, 2, 3; red) and stalk (location 4; black). Changes in $[Ca^{2+}]$ were expressed as $\Delta G/R$ (see Methods). **b** Bar graphs (mean ± SEM) summarize amplitude, 10–90% rise time, decay time ($\tau_{decay}$), and integral (amplitude × $\tau_{decay}$) of $Ca^{2+}$ transients in swellings (red; $n = 9/8$) and stalks (black; $n = 9/8$) at 100–300 Hz stimulation frequencies. Two-way repeated measures ANOVA with Bonferroni post hoc analyses was used to test significance. Critical $p$ values were corrected by the number (3) of comparisons (*$p < 0.0166$, ***$p < 0.0003$). See Supplementary Table 3. Scale bar: 2 μm (**a**)

of $Ca^{2+}$ transients varied approximately linearly with the stimulation frequency, we estimated that the $[Ca^{2+}]$-s were around or below the $K_d$ of the dye[58]. The larger $Ca^{2+}$ accumulation in swellings than stalks is likely the result of the larger surface to volume ratio and VGCC density in swellings as extrapolated from $I_{Ca2+}$ density recordings (Fig. 3 and Supplementary Figure 4). We suggest that the larger $Ca^{2+}$ accumulation in swellings may allow recruit distal SVs into the RRP during the late phase of or after the high-frequency train to expand the size of RRP, consequentially increasing the steady-state EPSCs to ensure the fidelity of postsynaptic spiking associated with complex calyces.

**Modeling functional diversity by distinct release modules.** We next examined whether a variable combination of SVs localized at shorter and longer coupling distances present in stalks and swellings, respectively, could explain the variability in the observed EGTA sensitivity. Because of the higher EGTA sensitivity of complex calyces (Fig. 5), we assumed that all the SVs in the swellings were more sensitive to EGTA. We considered a VGCC-SV coupling distance model to account for the presence or absence of the molecular spacer, septin. Thus we calculated the inhibition within the whole calyx (EGTA/control) as a weighted mean of the inhibitions of different coupled SV populations (Fig. 9a);

$$\frac{RRP_{St} \times Inh_{St} + RRP_{Sw} \times Inh_{Sw} \times n_{Sw}}{RRP_{St} + RRP_{Sw} \times n_{Sw}}, \quad (3)$$

where $n_{Sw}$ is the number of swellings, $RRP_{St}$ and $RRP_{Sw}$ are the RRP sizes in the entire stalk and RRP per swelling. Because RRP size in the whole calyx depends linearly on the number of swellings, $RRP_{St}$ and $RRP_{Sw}$ were estimated from the y-intercept and slope of the linear fit, respectively (Fig. 6c). As the number of swellings increased, the occupancy of tightly coupled SVs becomes reduced (Fig. 9a). To estimate the inhibitory effect of EGTA in stalks ($Inh_{St}$) and swellings ($Inh_{Sw}$) respectively, we fitted the data plot of EGTA inhibition using Equation (3). The best fit, in which the least squares mean was smallest (Fig. 9b),

was obtained when EGTA blocked release by 12% in stalks ($Inh_{St} = 0.88$) and by 56% in swellings ($Inh_{Sw} = 0.44$).

We next used numerical simulations of $Ca^{2+}$ reaction-diffusion and binding to a release sensor based on a perimeter release model[27] to estimate the coupling distance needed to reproduce different EGTA sensitivities and Pr. We first simulated $Ca^{2+}$ concentration change and Pr at various distances from a perimeter of VGCC cluster in control and in the presence of EGTA (see Methods). By taking the ratio of the two sets of Pr values we constructed the calibration curves for EGTA inhibition of release for a variety of open VGCC numbers, ranging from 1 to 20 channels (Fig. 9c). On the calibration curves, the 12% inhibition in stalks corresponds to a perimeter coupling distance of 14 nm regardless the number of open VGCCs in cluster (Fig. 9c, d). The 56% inhibition in swellings varied from 30 to 50 nm depending on the open channel number. These simulations confirm that the diversity of synaptic function within the population is likely created by building different ensembles of two distinct morphological modules in the nerve terminals: the stalk modules with shorter VGCC-SV coupling distance and higher Pr versus the swelling modules with longer VGCC-SV coupling distance and lower Pr. Increasing the number of swellings lowers the global Pr of the whole terminal, and establishes a morpho-functional continuum.

**Discussion**

This work and others demonstrate that VGCCs are organized in clusters at the presynaptic membrane and the number of VGCCs within the cluster is a crucial factor influencing Pr at an AZ[14,26,27,59]. In immature calyx, Sheng et al[26]. has directly measured the number of VGCCs, SV release and Pr at the same AZ using $I_{Ca2+}$ and capacitance measurements, and concluded that the number of VGCCs determines Pr of release-ready SVs. Large variability in the number of VGCCs per cluster (one cluster per AZ) has been uncovered in immature and mature calyces implicating that cluster size diversifies Pr at individual AZs[26,27]. We investigated VGCCs in distinct morphological modules of the mature calyx, and found stalks use larger VGCC clusters to

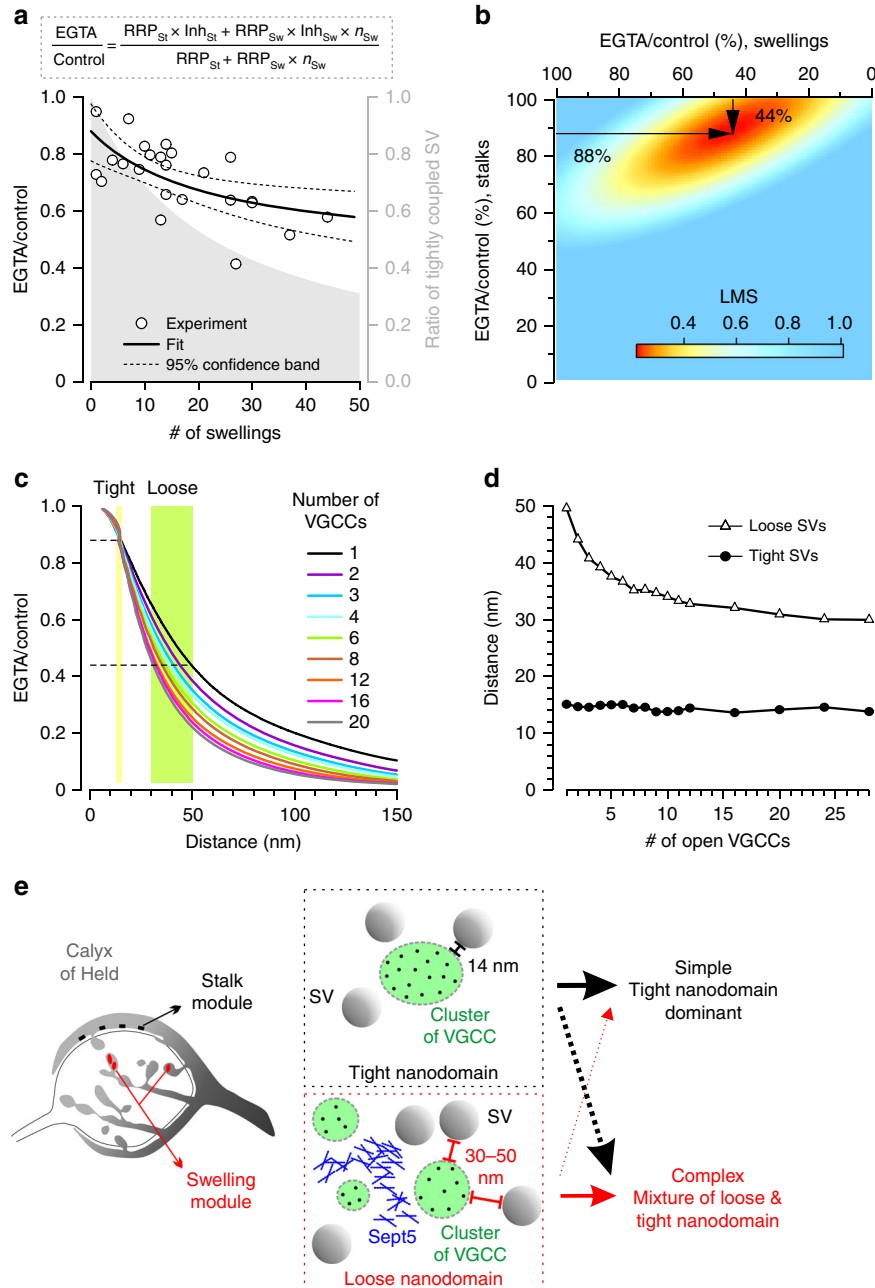

**Fig. 9** Numerical simulation of the VGCC-SV coupling distance reveals subpools of synaptic vesicles. **a** EGTA effect on EPSC amplitudes (EGTA/control) was plotted against the number of swellings (circles) and fitted (black) with the equation ± 95% confidence interval (dashed; see Equation (3) in Results). $n_{Sw}$ is the number of swellings, $RRP_{St}$ and $RRP_{Sw}$ are the RRP (readily releasable pool) sizes in entire stalks and per swelling, $Inh_{St}$ and $Inh_{Sw}$ are the inhibitory effects of EGTA in stalks and swellings, respectively. The gray zone (right axis) exhibits the occupancy of tightly coupled vesicles. **b** The least mean squares (LMS) of the fit exhibit the best EGTA effect in swellings (X-ordinate) and stalks (Y-ordinate). **c** The magnitude of inhibition by EGTA is a function of distance between vesicular $Ca^{2+}$ sensor and voltage-gated $Ca^{2+}$ channel (VGCC) cluster perimeter. Horizontal dashed lines indicate the EGTA inhibition in stalks (0.88) and in swellings (0.44). Vertical shaded regions indicate the range of distances between the sensor location and nearest open VGCC matching EGTA inhibition for different number of VGCCs in cluster. **d** Estimated coupling distance of loose and tight SVs were plotted against the number of VGCCs in cluster. **e** We describe a modular model of the heterogeneity in synaptic function. Variable proportions of distinct morphological modules (i.e. swelling and stalk) define functional heterogeneity. The stalk module (black) contains synaptic vesicles (SVs) with tighter coupling (14 nm; tight nanodomain) to larger VGCC clusters present in lower density. The swelling module (red) contains SVs with looser coupling (30–50 nm; loose nanodomain) to smaller clusters present in higher density. Septin 5 (Sept5) differentiates the coupling distances. Simple calyces mostly contain stalk modules while complex calyces are composed of a mixture of stalk and swelling modules. Arrows represent the contribution of the modules

elevate Pr and swellings use smaller VGCC clusters to lower Pr, such that heterogeneity in morphological complexity diversifies whole terminal Pr. We demonstrated that the $I_{Ca2+}$ density, larger in complex calyces, is not responsible for the lower whole terminal Pr, but may be associated with the larger number of release

sites using smaller VGCC clusters in swellings. Indeed, $I_{Ca2+}$ density is not necessarily predictive for Pr because the nanoscale organization of VGCCs determines the $Ca^{2+}$ nanodomains[27].

VGCCs are notoriously difficult to study in central synapses, for example, SDS-FRL technique gives the best resolution, but is

limited by the efficacy and specificity of antibody labeling as well as randomness of fracture points which make the 3-dimensional (3D) mapping of VGCC clusters onto the intact nerve terminals and their compartments impossible. Super-resolution microscopy is often limited by photobleaching, availability and efficacy of specific probes, and encounters difficulties in visualizing extensive structures in tissues, like brain slices[60]. By taking advantage of a knock-in mouse line, in which VGCCs were tagged with citrine, and a morphological volume tracer, we have determined the total number of clusters per terminals, and mapped out the topography of VGCC clusters in different presynaptic modules (i.e. swellings and stalks). Because of the resolution of confocal microscopy supported with Huygens deconvolution[61] yields lower resolution (~120 nm) than the electron or super-resolution microscopy, we provided the cumulative fluorescence of citrine tagged VGCCs within a cluster, a correlate of the number of VGCCs, rather than the size of the clusters. Furthermore, using a comparative measure of the uncalibrated citrine fluorescence intensities in the morphological modules we described fewer VGCCs per cluster and thus lower Pr in swellings than stalks in line with the perimeter release model[27]. Moreover, we found higher number of VGCC clusters per contact area on swellings indicating that the nanoscale geometry of clusters is more relevant to Pr than the number of clusters per contact area.

VGCC-SV coupling is essential to convey ultrafast signals, like sound localization cues, with a high temporal precision[62,63]. Thus we postulated that the heterogeneity in VGCC-SV coupling distance is a regulator of synaptic diversity, in addition to the number of channels per cluster. We tested this with a short EGTA injection into the calyx. We found that EGTA is more effective in attenuating release from complex than simple calyces, indicating longer coupling distance in swellings than stalks. To explore the molecular mechanism we tested the role of Sept5, a known down-regulator of synaptic strength by binding to syntaxin-1, where SNAP25 and VAMP also bind, and by extending the coupling distance[49,50]. Selective strengthening of complex calyces after Sept5 antibody injection supported the idea of longer coupling distance in swellings than stalks and highlighted Sept5 as a molecular substrate that contributes to the heterogeneity of synaptic transmission.

Based on the extent of EGTA effects on EPSCs, estimation of VGCC number and distributions, and local $Ca^{2+}$ transients, Nakamura et al[27]. proposed the perimeter model of neurotransmitter release that reproduced synaptic functions, and made specific predictions on the VGCC-SV coupling distance for a single AP. Using this formalism, we estimated that the coupling distance is tighter in stalk (~14 nm) and looser (30–50 nm) in swellings consistent with earlier estimation of whole-terminal coupling distance (~20 nm at P14[27,51]), which turns out to be similar to the geometric average of release from stalk and swellings. Because the average VGCC cluster occupies about 12% of the average AZ area and the typical AZ diameter is ~300 nm, we conclude that both the close and distally coupled SVs with ~50 nm diameter can be localized within the AZ[27,33]. As EGTA combined with numerical simulations indirectly tests the VGCC-SV coupling distance, we speculate that single AZs consist only one coupling modality and are localized exclusively to distinct morphological modules. However, we cannot exclude the possibility that the two coupling modalities coexist in subdomains of the same AZ[64] but in varying proportions in both morphological modules. Note that the effect of EGTA is much less potent in these mature synapses than immature terminals (up to 80% inhibition), which primarily employ microdomain release modality with a longer coupling distance as we originally defined (>60 nm; [17,28,48,65]).

Swellings are likely the primary sites for RRP expansion[20] because they accumulate $Ca^{2+}$ more than stalks as a result of larger $I_{Ca2+}$ density and compartmentalization. Larger $Ca^{2+}$ accumulation in swellings may drive $Ca^{2+}$-dependent replenishment of SVs, which, in turn, boosts the size of RRP in complex calyces[31,57,66,67]. In parallel, smaller VGCC clusters and longer coupling distances on swellings keeps the initial Pr low, but help sustain the fidelity of neurotransmission as $Ca^{2+}$ builds up during repetitive activity.

Chen et al[51]. demonstrated broad distribution in VGCC-SV distances and suggested two subpools of RRP-SVs: one tightly coupled, fast-releasing at ~15 nm distance, and another relatively loosely coupled slowly releasing pool between 25–150 nm distances. Single APs are unlikely able to release SVs from distances larger than 50 nm[51]. Consistent with the presence of two subpools of SVs at various distances from VGCCs in distinct morphological modules, we found larger EGTA effect in swelling-rich complex calyces. We can rationalize that the two subpools may be present respectively in stalk and swelling modules, but varying the proportion of two modules can give rise to the differences in EGTA sensitivity. Alternatively, the two subpools are present in both morphological modules but in different proportion. We therefore describe here a subpool of SVs that are located at longer distances (30–50 nm), which are regulated in their number and contribute to the overall heterogeneity of global synaptic efficacy, RRP size and STP across calyx terminals.

The calyx of Held synapse converts ultrafast, contralateral, excitatory inputs to inhibitory outputs onto neurons in the medial and lateral superior olive, where precisely timed ipsilateral, excitatory inputs also innervate, and code the interaural time or level difference (ITD or ILD) of sound arriving at two ears[18]. Temporal precision and reliability of information transfer across a wide spectrum of sensory stimuli is critical to ITD and ILD computation. Precision and reliability at high frequency transmission is achieved by the trade-off between Pr and RRP across the calyx population. Calyces use variable proportions of two morphological modules containing different SV-VGCC topographies to achieve the trade-off. The swelling module contains larger number of VGCCs organized into smaller clusters, coupled more loosely to SVs with lower Pr, versus the stalk module with sparse but larger clusters of VGCCs, coupled more tightly to SVs with higher Pr (i.e. tight [14 nm] and loose [30–50 nm] coupling). The two discrete coupling modalities are differentiated by Sept5 (Fig. 9e). Tuning the number of swelling modules inversely diversifies the whole-terminal Pr but directly the RRP, and establishes the morpho-functional continuum (Figs. 1b, 4g and 9e). RRP varies directly with calyx complexity by two mechanisms. Adding more swellings to stalks increases the number of AZs, and swellings accumulating $Ca^{2+}$ more than stalks boost the RRP likely by accelerating the replenishment of SVs. Temporal precision is therefore achieved by employing stalks with tightly coupled SVs and higher Pr to drive early spike initiation in postsynaptic neurons[23,25,27], while addition of swellings with more loosely coupled SVs lowers initial Pr and expands RRP to increase the steady-state synaptic transmission, thus endowing synapses with reliability during high-frequency transmission optimal for their spatiotemporal roles.

Our findings at calyx can be generalized to other neural circuits in which postsynaptic neurons are innervated by multiple inputs with heterogeneous Pr-s. The weighted average of Pr across release sites determines the fidelity of spike timing and number, both of which are important cues for computation. By varying the ensembles of morpho-functional modules of nerve terminals, a single population of ultrafast central synapses can be tuned to convey wide spectrum of sensory information at different intensities with high temporal precision.

## Methods

**Slice preparation.** Acute transverse brainstem slices (250 μm) containing medial nucleus of trapezoid body (MNTB) were prepared from P16–21 CD1/C57 mice of either sex[16] according to a protocol approved by The Hospital for Sick Children Animal Care Committee. Mice were housed in the facility certified by the Canadian Council on Animal Care. After decapitation the brain was placed in ice-cold cutting solution containing (in mM): 125 NaCl, 2.5 KCl, 10 glucose, 1.25 $NaH_2PO_4$, 2 Na-pyruvate, 3 myo-inositol, 0.5 ascorbic acid, 26 $NaHCO_3$, 3 $MgCl_2$, and 0.1 $CaCl_2$ (pH = 7.4) and sliced on a vibratome (VT1200S, Leica). Slices were transferred into artificial cerebrospinal fluid (aCSF, same composition as cutting solution except in mM, 1 $MgCl_2$, 2 $CaCl_2$, pH = 7.4), incubated for 1 h at 35 °C then stored at room temperature until use. Solutions were oxygenated with 95% $O_2$ and 5% $CO_2$.

**Electrophysiology.** The auditory brainstem slices were continuously perfused (1–2 ml/min) in the recording chamber at room temperature (22 °C) with aCSF supplemented with bicuculline (10 μM) and strychnine (1 μM) to block inhibitory inputs. Principal cells embraced by the calyx of Held synapses in the MNTB were visualized by an upright microscope (Axio Examiner, Carl Zeiss) equipped with a CCD monochrome video camera (IR-1000, DAGE-MTI) and identified by their localization, shape and size. In vivo, mature calyces are heterogeneous in morphology and spiking fidelity recapitulating the diversity observed in slices at room temperature[22,68]. Previously, we tested how physiological temperature (35 °C) impact spiking fidelity and found that the heterogeneity reflected in spiking onset and frequency is fundamentally similar between the two conditions but emerges at higher frequencies at physiological temperature[4,21]. Nonetheless, the observed temperature-dependence of SV release and supply encompasses different mechanism, e.g. ultrafast endocytosis is observed at physiological temperature and regulated by dynamin and actin while endocytosis at room temperature is clathrin-dependent[69,70]. Electrophysiology data were sampled (50 kHz) and low-pass filtered (4 kHz four-pole Bessel filter) by a Multiclamp 700B dual channel amplifier (Molecular Devices).

To test the effect of presynaptic EGTA or SP20 antibody (1:1000, Mouse-anti-Septin 5, 200 μg/ml; Santa Cruz Biotech, Cat# sc-65512, Lot# K1407, RRID: AB_1129364) injection on synaptic transmission, we performed whole-cell voltage-clamp recordings of the excitatory postsynaptic currents (EPSC) from the principle cells. 2.5–3 MΩ pipettes were pulled (PP-830, Narishige) from borosilicate glass with filament (WPI) and filled with (in mM) 97.5 K-gluconate, 32.5 CsCl, 5 EGTA, 10 HEPES, 1 $MgCl_2$, 30 tetraethylammonium (TEA), and 3 lidocaine N-ethyl bromide (QX-314, Br-), and pH = 7.2 was set by KOH (about 310 mOsm). Holding potential was kept at −60 mV. Series resistance ranged between 4 and 8 MΩ and was compensated to 90%. To evoke EPSCs afferent to calyx terminals were stimulated at the midline (<25 V, 0.2 ms; 30–50% above threshold to avoid presynaptic AP failures) using a Master 8 stimulator (A.M.P.I.) coupled to a bipolar platinum electrode. Meanwhile, using a second patch-clamp electrode (6.5–8 MΩ) filled with (in mM) 97.5 K-gluconate, 32.5 KCl, 10 EGTA or 0.5 EGTA plus SP20 antibody, 40 HEPES, 1 $MgCl_2$, 2 Na-ATP, 0.5 Na-GTP, 12 phosphocreatine di(tris) salt and 0.06 Alexa Fluor 594 hydrazide (Invitrogen), pH adjusted to 7.3 with KOH (about 330 mOsm), we carried out cell-attached voltage-clamp recordings (>1 GΩ) on the calyces to control the fidelity of presynaptic action potential (AP) initiation by recording the presynaptic compound AP currents (pre-$I_{AP}$). Cell-attached recording allowed us to minimally disturb the presynaptic terminal and keep the calyx structure unrevealed until the recordings from postsynaptic neurons were complete. EPSCs were recorded during 300 Hz test trains (200 ms). Each test train was separated by at least 45 s to avoid carryover of short-term synaptic plasticity between tests. After the control test trains, the presynaptic patch membrane was ruptured and the calyces were loaded with EGTA (10 mM) or SP20 for 3–5 min via diffusion. The series resistance was <30 MΩ. The efficiency of loading was always tested by imaging the Alexa 594 labeled calyces (see details below). The calyx terminals were kept at -80 mV holding potential during the EGTA injection. After the loading period the presynaptic electrodes were carefully withdrawn to allow resealing (>1 GΩ) of the presynaptic membrane, and the test trains were repeated to record EPSCs in the presence of EGTA. Potentials were not corrected for liquid junction potential. Miniature EPSCs were recorded in a separate set of experiments. Previously, we have verified the specificity and potency of the SP20 antibody by injecting it into Chinese hamster ovary (CHO) cells expressing Sept5-GFP or actin-GFP filaments and into calyces from Sept5 wild-type and knock-out mice and demonstrated that the antibody is a specific and potent functional inhibitor of Sept5 filaments in the calyx of Held synapses[50].

The effect of EGTA on spike fidelity was tested in dual cell-attached voltage-clamp configuration (>1 GΩ) while the afferent axons were stimulated at the midline (300 Hz, 500 ms; for details see above). We kept the postsynaptic and presynaptic membrane potential at −60 and −80 mV, respectively. The postsynaptic electrode (6–8 MΩ) contained (in mM) 97.5 K-gluconate, 32.5 KCl, 0.5 EGTA, 10 HEPES, 1 $MgCl_2$, and pH = 7.2 was set by KOH. We recorded the presynaptic fidelity of spiking with a presynaptic electrode (6.5–8 MΩ) filled with (in mM) 97.5 K-gluconate, 32.5 KCl, 10 EGTA, 40 HEPES, 1 $MgCl_2$, 2 Na-ATP, 0.5 Na-GTP, 12 phosphocreatine di(tris) salt, and 0.06 Alexa Fluor 594 hydrazide, pH adjusted to 7.3 with KOH. After testing the baseline postsynaptic fidelity we broke

the presynaptic patch membrane, loaded the terminal with EGTA for 3–5 min, then gently removed the presynaptic electrode to reseal the membrane (>1 GΩ) and tested the postsynaptic fidelity again.

For presynaptic $Ca^{2+}$ current recordings ($I_{Ca2+}$) the patch electrodes with resistance of 5–6 MΩ were filled with (in mM) 110 CsCl, 0.5 EGTA, 40 HEPES, 1 $MgCl_2$, 2 Na-ATP, 0.5 Na-GTP, 12 phosphocreatine d(tris) salt, 30 TEA and 0.06 Alexa Fluor 594 hydrazide (Invitrogen), pH adjusted to 7.3 with CsOH (about 330 mOsm). In addition to bicuculline (10 μM) and strychnine (1 μM) we included tetrodotoxin (TTX, 1 μM), TEA (10 mM) and 4-aminopyridine (4-AP, 0.3 mM) into the aCSF to block $Na^+$ and $K^+$ channels. The extracellular $Ca^{2+}$ and $Mg^{2+}$ concentrations were 2 and 1 mM, respectively. The presynaptic series resistance was <10 MΩ and compensated to 90%. Capacitive current and leak was subtracted online with the P/4 protocol. Voltage commands applied to test $I_{Ca2+}$ are detailed in the text.

To test the propagation of APs into swellings and stalks compound action potential currents (pre-$I_{AP}$) were recorded in cell-attached configuration at multiple locations (≥2) on each calyx. Some pre-$I_{AP}$s were recorded sequentially others simultaneously (dual recordings) at different sites. In sequential recordings labeling and imaging after the first patch (A594 loading in whole-cell mode and TPLSM z-stack image) helped identify and target the small compartments. The data derived either sequentially or simultaneously were pooled together. APs were evoked by extracellular midline axonal stimulation. 6.5–8 MΩ pipettes were filled with (in mM) 97.5 K-gluconate, 32.5 KCl, 0.5 EGTA, 40 HEPES, 1 $MgCl_2$, 2 Na-ATP, 0.5 Na-GTP, 12 phosphocreatine di(tris) salt and 0.06 Alexa Fluor 594 hydrazide (Invitrogen), pH adjusted to 7.3 with KOH (about 330 mOsm). On every calyx we formed at least one >1 GΩ seal resistance patch which was opened to label the terminal with Alexa 594. The other patches were often loose cell-attached (>50 MΩ). The calyx morphology was always registered by two-photon laser scanning microscopy (see below).

Data were analyzed offline using the pCLAMP 10 software package (Molecular Devices), MiniAnalysis (Synaptosoft), Excel (Microsoft) and Igor Pro (WaveMetrics) equipped with Neuromatic module. Individual EPSC trains were baseline subtracted and averaged. We quantified the EPSC amplitudes recorded during the stimulus trains (300 Hz, 200 ms). The EPSC amplitudes after the facilitation were fitted with a double-exponential function to obtain the time constants and amplitude components of the depletion of SV subpools ($\tau_{decay1}$, $\tau_{decay2}$, $A_1$, $A_2$, respectively). In cases where we noted a slight deviation of EPSC amplitudes at steady-state depression from a straight line likely reflecting the $Ca^{2+}$-dependent augmentation of SV replenishment (Fig. 1a;[31]) and this deviation visibly affected the quality of fitting, we fitted the two-exponential function to a shorter segment of the train (after facilitation but before the deviation). Paired-pulse ratios (PPR) were calculated by dividing the second EPSC amplitude by the first during 300 Hz stimulus trains (EPSC2/EPSC1). The size of the readily releasable pool (RRP) of synaptic vesicles was estimated from the depleting stimulus train (300 Hz, 200 ms) by back-extrapolating to time 0 ms from last 50 ms of the steady-state portion of the cumulative EPSC amplitude curves[29]. We also quantified RRP size applying the method described by Wesseling and Lo[30] which considers the variable rate at which SVs are recruited into the RRP during a high-frequency train and minimizes the counting of quanta from reserve pools into the RRP:

$$\mathrm{fe} = \frac{r(1)}{r(\infty)} * \left(1 - e^{\frac{-a}{\nu}}\right) \quad (4)$$

$$\mathrm{fe} = \frac{r(1)}{\sum_{i=1}^{S} r(i) * e^{\frac{-a(s-i)}{\nu}}}, \quad (5)$$

where fe is the fusion efficiency (fraction of RRP released by the first EPSC), $r(1)$ is the amplitude of the first EPSC, $r(\infty)$ is the amplitude of the steady-state EPSC, $\alpha$ is the rate of pool filling, $S$ is the number of stimuli in the train, and $\nu$ is the frequency of stimulation. The following equation can then derive the total current from the readily releasable synaptic vesicles:

$$N = \sum_{i=1}^{S} r(i) - \omega(S), \quad (6)$$

where $\omega(S)$ is the total current released from the newly available SV pool during the train, $N$ is the total current from RRP. To estimate the release probability (Pr), we divided the amplitude of the first EPSC from the 300 Hz trains by the size of the RRP. We defined the steady-state EPSC amplitude (EPSC$_{ss}$) as the average of EPSC amplitudes during the last 50 ms of the depleting train. RRP size estimation by linearly fitting the 100–150 ms segment at the steady-state portion of cumulative EPSCs but before the activation of $Ca^{2+}$-dependent replenishment of SVs (Fig. 1a) revealed <4% error in control RRP and Pr of simple and complex calyces. The propagated error estimated by the effect of ± standard deviation of a previously derived fit parameter on a subsequently fitted mean curve (Figs. 4g, 9a) was <4%. Peak-to-peak time of pre-$I_{AP}$, an approximate of the AP half-width, was defined as the interval between the first inward, and the second outward peak (peak$_1$ and peak$_2$) of the current attributed to the maximal rate of AP depolarization and repolarization, respectively (Fig. 2a;[38,41,71]). We calculated the synaptic delay as

the delay between peak$_1$ or peak$_2$ of pre-I$_{AP}$ and onset of EPSC (SD$_1$ and SD$_2$, respectively; Fig. 5a). Although the synaptic delays after EGTA injection were not measured because the presynaptic pipette was removed, the rise and peak time of evoked EPSCs align well with those before EGTA, suggesting little changes in this parameter. Fidelity of postsynaptic spiking was defined as the ratio of the number of successful postsynaptic compound action potential currents (post-I$_{AP}$) and the number of stimuli. Ca$^{2+}$ current density (I$_{Ca2+ dens}$) was calculated by dividing the maximal step or tail Ca$^{2+}$ current (I$_{Ca2+ max}$, defined as the difference between the peak of the largest I$_{Ca2+}$ and the baseline before the voltage step) by the whole-cell capacitance (C$_{whole-cell}$).

**Calcium imaging.** Calcium imaging was performed using 810 nm pulsed laser light generated by Chameleon Ultra Ti:Sapphire laser system (Verdi and VPUF laser head; Coherent) and TPLSM (LSM 710 NLO; Carl Zeiss). The laser beam was scanned across the sample through a ×63 water-immersion objective (1.0 NA, W Plan-Apochromat, Carl Zeiss). For recording single AP-evoked Ca$^{2+}$ transient the pipette (6.5–8 MΩ) was filled with (in mM) 97.5 K-gluconate, 32.5 KCl, 40 HEPES, 1 MgCl$_2$, 2 Na-ATP, 0.5 Na-GTP, 12 phosphocreatine di(tris) salt, 0.05 Fluo-4 ($K_D$ = 345 nM; Invitrogen) and 0.015 Alexa Fluor 594 hydrazide (Invitrogen), pH adjusted to 7.3 with KOH (about 330 mOsm). To record AP train-evoked Ca$^{2+}$ transient Fluo-4 was replaced with the low affinity Ca$^{2+}$ indicator Fluo-4FF ($K_D$ = 9.7 μM). ATP, GTP, and phosphocreatine di(tris) salt were omitted from the pipette in these experiments. Calyces were kept at −80 mV and dialyzed for >15 min APs were delivered either after pipette removal (>1 GΩ) by extracellular axonal stimulation (see above) or in current-clamp mode by current injection (up to 3000 pA, 0.2 ms). The emitted green and red fluorescence signals were spectrally separated using appropriate combinations of filters (BP500–550, BP600–660) and a dichroic mirror (MBS 760+) and detected by non-descanned detectors (NDD). Excitation laser light was always kept as low as required to attain a sufficient signal-to-noise ratio and minimal photo-damage. We used line-scan mode (0.3–0.5 kHz) to image single AP-evoked Ca$^{2+}$ transients and frame scan mode (30–60 Hz) for AP train-evoked Ca$^{2+}$ transients. To obtain a sufficient signal-to-noise ratio 10–35 or 5–10 sweeps were averaged for single APs or AP trains, respectively.

Imaging data were analyzed using Zen 2009 (Carl Zeiss), Excel, Igor Pro Neuromatic package and ImageJ (http://imagej.nih.gov/ij/, NIH[72]) softwares. Both green and red pixel intensities within a polygonal region of interest were averaged for each frame or line. To avoid problems with background correction[73] due to the proximity of seal formation and the site of imaging we were very careful to prevent dye ejection into the extracellular space (fast sealing, only slightly increased pipette pressure during seal formation), recordings with increased background fluorescence were excluded from the study and background correction was not performed. ΔG over R ratio (ΔG/R) was calculated for each frame or line to follow changes in [Ca$^{2+}$] and was defined as $(G–G_{basal})/R_{basal}$ where G is the actual average fluorescence of a spatial extent of a swelling or a stalk, and $G_{basal}$ and $R_{basal}$ are the baseline fluorescence in the green and red channel measured 50–100 ms before the stimuli. The amplitude of single AP-evoked Ca$^{2+}$ transient was calculated as an average of a 5 ms window around the peak of ΔG/R signal. The decays of Ca$^{2+}$ transients were estimated by a single exponential fit. In some cases multiple swellings and/or stalks were recorded from a single calyx. To avoid the overrepresentation of a single calyx in the statistics, first, we averaged the swelling and/or stalk data for every calyx and then we used these values in the statistics.

**Morphological analyses.** High-resolution z-stack images of calyces (0.5 μm steps) were acquired right after electrophysiology recording and Ca$^{2+}$ imaging using the same TPLSM (see above) to obtain the 3D structures labeled with Alexa 594. Zen and ImageJ softwares were used to render 3D reconstructions and to adjust contrast and brightness. Calyces that had low signal intensity or appeared damaged were not included in the study. Elliptical, small structures connected to the larger structures (stalk) via one or more thin and short neck(s) were considered swellings. As we have shown[4] that the number of swellings correlates with other morphological parameters describing the complexity of calyces (surface area, volume, and perimeter), we used the number of swellings for calyx classification. The number of swellings on each calyx was determined by visual inspection of 3D images. Calyces with ≤10 swellings were considered simple in morphology and complex if they carried >20 (Fig. 1a).

**Anterograde tracing and VGCC cluster quantification.** We pulled 7–8 MΩ pipettes from borosilicate glass capillaries (WPI), coated the tips with dextran conjugated Alexa Fluor 594 (10,000 MW, dissolved in 1.5% bovine serum albumin), and inserted them into the midline of acute transverse brainstem slices[74] prepared from a knock-in mouse line (C57 background) in which the N-terminus of P/Q-type voltage-gated Ca$^{2+}$ channel (VGCC) α$_1$ subunit was tagged with citrine, a GFP variant (Cacna1a$^{Citrine}$, [46]). We held the tips in place until the dry crystals of the dye dissolved. The slices were incubated for up to 60–90 min at 24 °C which allowed the labeling of axons giving rise to calyx of Held terminals and the post-fixation identification of the morphological modules (swelling and stalk). Slices were fixed in 4% paraformaldehyde in PBS for 1 h at room temperature then rinsed in PBS thoroughly, mounted (Prolong Diamond Antifade Mountant, Invitrogen), dried and sealed with coverslip (1.5 H, Marienfeld-Superior). To quantify VGCC clusters we acquired z-stack images (0.3 μm steps) of calyces by a Leica TCS

SP8 Confocal imaging system equipped with Leica DMI6000 inverted microscope and super-sensitive hybrid detectors (HyD). The system was controlled by the Leica LAS X software. 515 and 598 nm laser beam generated by White Light Laser was sequentially scanned across the sample through a ×63 oil-immersion objective (1.4 NA, pixel spacing x,y = 51.2 ± 2.0 nm, n = 25). We separated the dye and autofluorescence using the fluorescence life time gating function (0.3–6.0 ns). Image quality of citrine labeling was improved by Huygens deconvolution within the Leica software package (SVI software). Citrine-tagged P/Q-type VGCC clusters on each calyx were quantified by the 3D object counter plugin in ImageJ ([72,75]; https://imagej.net/3D_Objects_Counter, online manual available, NIH). The citrine fluorescence throughout the entire calyx was thresholded above the background level then every identified cluster was marked with a unique number deriving the total number of clusters per calyx. To explore the distribution of VGCC clusters and number of VGCCs per cluster within the two morphological modules, we identified the swellings and stalks using the A594 dextran labeling, and paired the clusters on swellings and stalks located in equal depths from the slice surface with a clear top view to avoid variation in excitation power and emission detection efficiency due to tissue scattering. We quantified the contact area using the region of interest manager tool (ROI, ImageJ) and calculated the number of clusters per contact area on swellings and stalks for each calyx as the sum of clusters on all swellings and stalks divided by the sum of contact areas of all swellings and stalks, respectively (Excel, Microsoft; Fig. 4e). To compare the number of VGCCs per cluster within the paired morphological modules we took advantage of the 100% labeling efficiency of the knock-in mice and quantified the cumulative fluorescence intensities per cluster in swelling and stalk (or the sum of fluorescence intensities of all voxels in the cluster). Data represent the average of all clusters within the paired morphological modules of a given calyx. Since the citrine fluorescence intensities were uncalibrated, we expressed the difference between the morphological modules as relative values (stalk/swelling, Fig. 4f).

**Numerical simulations.** Ca$^{2+}$ entry, diffusion and binding with buffer were simulated by numerically integrating partial differential equations applying an explicit finite-element (Euler) method, using a Java-based 3D diffusion-reaction simulator (D3D) running on a Windows 7 operating system[27]. We modeled only open VGCCs during an AP. The time course of Ca$^{2+}$ influx through a channel was adopted from whole-terminal Ca$^{2+}$ current evoked by P16–18 AP-waveform voltage command (see Fig. 6b in ref. [38]). The half-duration of the Ca$^{2+}$ current was 0.26 ms. The peak amplitude of single channel current was set at 0.35 pA[27], which is calculated from single channel conductance[26] corrected for 2 mM Ca$^{2+}$ and driving force of Ca$^{2+}$ during AP. VGCC cluster composed of such open VGCCs was placed at the center of simulation volume (500 nm in x-y, 1000 nm in z). Channel to channel distance was 35 nm, derived from SDS-digested freeze fracture replica labeling for Cav2.1[27]. We used simulation voxels of 5 × 5 × 5 nm. Model parameters were set to match experimental condition: resting free [Ca$^{2+}$] = 50 nM, Ca$^{2+}$ diffusion coefficient ($D_{Ca2+}$) = 220 μm$^2$/s [76], endogenous fixed buffer (EFB) binding ratio = 40 [42], EFB concentration = 4 mM, EFB $K_D$ = 100 μM, and an EFB forward binding rate of ($k_{on}$) = 1 × 10$^8$/Msec [27]. 1.1 mM free ATP was included in simulation to match the expected free ATP concentration in our patch pipette given total [ATP] and total [Mg$^{2+}$](calculated using Max Chelator). The values for Ca$^{2+}$ binding to ATP were as follows: $K_{D, Ca}$ = 200 μM, $k_{on, Ca}$ = 5 × 10$^8$/Msec [77], $D_{ATP}$ = 220 μm$^2$/sec. We assumed that ATP concentration was not changed before and after membrane rapture. The parameters for EGTA were as follows: $k_{on}$ = 1.05 × 10$^7$/Msec, $K_D$ = 70 nM, $D_{EGTA}$ = 220 μm$^2$/sec [78]. In simulation the EGTA concentration after membrane rupture was set at 7.5 mM, considering the time course of Ca$^{2+}$ indicator dye penetration to the calyx [42]. The time step of simulation was calculated according to a stability criterion, $h = (3 × D × dt)/δx^2$, where D is the diffusion coefficient of the fastest species and δx is the length of vertices of simulation voxels. Using simulated [Ca$^{2+}$] time course ([Ca$^{2+}$](t)) at each 5 nm simulation voxel, we next calculated vesicular release probability (Pr). This calculation was performed using a 5-site Ca$^{2+}$-dependent release model for mature mice[79] run by a forward Euler numerical integration routine (Igor Pro). [Ca$^{2+}$](t) and Pr were simulated both for control condition and in the presence of EGTA. By taking the ratio, the inhibitory effect of EGTA could be estimated in each simulation voxel. After depicting spatial map of the effectiveness of EGTA in the release face of simulation field, we created calibration curves of inhibition of EGTA corresponding to specific distance in 1 nm steps. In this way, we could predict the magnitude of inhibition by EGTA for specific open channel number and specific perimeter coupling distance.

**Statistics.** For statistical analyses of significance, paired t-test (p.t.t.), unpaired t-test (u.t.t.), one sample t-test with a hypothetical mean = 1 (1-s.t.t.), and one-way ordinary or two-way repeated measures ANOVA (analysis of variance) with Bonferroni post hoc test were used with two-tailed probabilities. We applied Kolmogorov and Smirnov method to test the Gaussian distribution of the samples. We used unpaired t-test with Welch correction if the samples had different standard deviations (u.t.t.W.). Averages were expressed as mean ± SEM (standard error of mean). Fitting results are shown as mean ± 95% confidence interval (CI). The n shows the number of experiments either as calyx/animals or morphological module/animals depending on the experiment. Differences were considered significant at *$p < 0.05$, **$p < 0.01$, and ***$p < 0.001$, except for the Bonferroni post hoc

test where the significance levels were corrected by the number of comparisons (see details in text). Degrees of freedom (df) was detailed for each statistics. The parameters of F distribution (degrees of freedom numerator (dfn) and denominator (dfd)), used to calculate probability values for ANOVA, were given as indices of the $F$ value ($F_{(dfn,dfd)}$). $R$ indicates Pearson's correlation coefficient. Statistics were calculated using Graph Pad Prizm 4.

**Materials**. AlexaFluor-594 hydrazide, AlexaFluor-594 dextran (10,000 MW), Fluo-4 and Fluo-4FF pentapotassium salts were purchased from Invitrogen. CsCl and NaHCO$_3$ were obtained from ACP Chemicals, tetrodotoxin (TTX) from Alomone Labs, N-ethyllidocaine bromide (QX-314, Br-) and bicuculline from Tocris Bioscience, SP20 antibody from Santa Cruz Biotechnology (Cat# sc-65512, Lot# K1407, RRID: AB_1129364), paraformaldehyde from Electron Microscopy Sciences. We ordered NaCl, KCl, glucose, NaH$_2$PO$_4$, Na-pyruvate, myo-inositol, L-ascorbic acid, MgCl$_2$, CaCl$_2$, K-gluconate, ethylene glycol-bis(2-aminoethylether)-N,N,N′,N′-tetraacetic acid (EGTA), HEPES, phosphocreatine, Na-ATP, Na-GTP, phosphocreatine di(tris) salt, tetraethylammonium chloride (TEA), 4-aminopyridine (4-AP) and Strychnine from Sigma-Aldrich.

**Code availability**. All relevant codes are available from the authors.

**Reporting Summary**. Further information on experimental design is available in the Nature Research Reporting Summary linked to this Article.

### Data availability
The data that support the findings of this study are available from the corresponding author upon reasonable request. A Reporting Summary for this Article is available as a Supplementary Information file.

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

## Acknowledgements

This work is supported by Operating Grants from the Canadian Institutes of Health Research (MOP-77610, MOP-81159, MOP-14692, VIH-105441), Natural Sciences and Engineering Research Council of Canada (Discovery Grant, RGPIN-2017-06665) and Canada Research Chair to L.Y.W.; D.A.D. is supported by the Centre National de la Recherche Scientifique, Fondation pour la Recherche Medicale and the Agence Nationale de la Recherche (ANR-2010-BLANC-1411 and ANR-13-BSV4-0016), and his laboratory is a member of the Bio-Psy Laboratory of Excellence. Y.N. was supported by the Japan Society for the Promotion of Science (KAKENHI Grant JP17K07064).

## Author contributions

A.F. and L.Y.W. designed the project. A.F., Y.M.Y., and L.Y.W. conducted the experiments. A.F. analyzed the data. Y.N. and D.A.D. performed the simulations. A.F., Y.N., D.A.D., and L.Y.W. wrote the manuscript. S.H. and M.D.M. provided material.

## Additional information

**Competing interests:** The authors declare no competing interests.

