## [Peer Review File · Nature Communications]

Reviewers' comments:

Reviewer #1 (Remarks to the Author):

The present MS describes experiments aiming to understand the mechanisms underlying the heterogeneity of functional properties among Calyx of Held synapses. The authors built on their previous finding that calyx synapses show substantial diversity and here they performed a series of physiology, modelling and anatomy experiments to demonstrate that different coupling distances underlie different release probabilities and that the functional diversity correlates with the structural heterogeneity of these synapses. The reviewer found the concept of creating heterogeneity by two distinct mechanisms and blending them in a variable degree interesting and novel. Despite this, there are few issues with the MS, which should be addressed.

The experiments with high-affinity Ca²⁺ indicators do not add anything to the MS. The authors themselves state that the differences in Pr cannot be explained by global Ca events, because that depends on the volume of the structures. Then the authors performed Ca²⁺ measurements with a low-affinity dye (Fluo4FF) to test nanodomain Ca²⁺ concentrations. However, for some reasons the authors used long trains of APs, the interpretation of which is very complicated given its dependence on diffusion, Ca buffering and potentially even extrusion. It is also unknown why the authors decided to illustrate this potentially more meaningful finding in the supplementary figure.

Despite their own argument for the lack of meaningful interpretation of the Ca transients in swellings and stalks, the authors start the next paragraph (page 9) that 'As we found that swellings are characterized by larger single AP-evoked Ca²⁺ transients than stalks, we reasoned that the total number and the density of VGCCs may be higher...'. This does not make sense! Only differences in the surface to volume ratio can explain the data.

The authors describe their measurements on cluster geometry as volume (3D structure). The reviewer is sceptical about the resolution of their technique to provide reliable estimate of the real cluster volume. It is also interesting why the authors decided to measure the 'volume' and not the 2D projection surface area? That would be much closer to the relevant morphological structure! The authors should definitely provide some evidence that their measure has anything to do with the AZ geometry!

The EGTA experiments indicate different coupling distances among Calyces with different number of swellings. The reviewer is puzzled about the top two sentences in page 13! The authors state that a shorter coupling distance should have shorter synaptic delay and EPSC time course, but their data is inconsistent with this. What does it mean?

It is much less informative to provide the ratios of two values (e.g. in fig. 5B) than providing both values in a pairwise plot. It is also true for the result section where the authors describe the effect of EGTA; they should tell that the value was reduced from X to Y rather than telling their ratios. And in few other places in the MS.

Reviewer #2 (Remarks to the Author):

The work by Fekete et al presents a large number of technically very difficult experiments that together strongly support the idea that release probability at the Calyx of Held is regulated by synaptic morphology, by the distribution of calcium channel and by the proximity of synaptic vesicles to calcium channels. At the Calyx of Held all vesicle release sites are formed by the very same presynaptic axon, making a particularly attractive target for studies on release site diversity. The authors describe two morphological 'modules': one with vesicles in close proximity to large calcium channel clusters and a high release probability; the other with fewer calcium channels, less tight coupling and lower release probability. They propose that the overall characteristics of a Calyx is determined by varying amounts of these two modules.

Overall, the data is convincing (see below) and while the writing could be clearer the authors explain the context and background of the experiments well. A bit more discussion on the physiological relevance of fine-tuning at this particular synapse would be helpful.

The major conclusions of the paper are not very novel and are hardly surprising. (I appreciate that I might have missed a crucial part in which case the writing is not as clear as I thought!) Very similar experiments and results have been published before by the same group and others. Confirmation is always good, especially when experiments are as difficult as these! One value of the present publication then lies in being a comprehensive compendium and as such will serve as a valuable one-stop reference. That said, the use of a citrin-P/Q knock-in to determine calcium channel density and spatial clustering is clever and very valuable.

The idea of two modules is interesting but clashes with the concept of the 'morpho-functional continuum' that the authors keep referring to. In fact, a duality only arises when the authors create two distinct and arbitrary sets (less than 10 'swellings' and >20 swellings; including the 10-20 data as a third set was previously done by the same others). I suggest that the authors more clearly formulate whether they want to treat this as a two-module or a continuum idea for this publication.

More detailed comments:

- One of my favorite parts of the paper is shown in Fig 4i. It took me a while to appreciate what was fit and what was derived, but the way I got to understand it was that the straight line in 4i is based on fitting data in 4g and 4h (grey) and then superimposing the result on to the data in 4i. This could probably be made a lot clearer than it currently is. Also, that the data in 4g-I is the same as in Fig1b must be indicated much more clearly. I also would have liked to see the straight line fit from 1b superimposed onto the model fit in 4i. But the fact that after so many divisions, normalizations and extrapolations the model superimposes onto the data is remarkable!

- Figure 4i also slightly inoculates against one of my main concerns about the data and how it is presented. Much of the data is derived from calculations, extrapolations and fits, all of which have variance associated with them. None of this variance is carried forward in any of the calculations or data presentations, leaving the reader at a loss of how well the data conforms to all the models and assumptions. In other words, how robust are the conclusions?

For example, the decrement in amplitude during a train is fit with two exponentials and the RRP is calculated from back-extrapolation to the first stimulus. As has been pointed out many times, this is clearly in error as it does not take replenishment of pools into account. Moreover, Figure 1a, complex shows amplitudes that first decrement to a low level, but then increase again by at least 50% for the last 100 ms. Back-extrapolating from the last 50ms thus clearly is not appropriate. Would a more complex model make a difference to the overall conclusion? I would think so, but it also might introduce yet another variable making sound conclusions and predictions even harder.

- While the authors have previously done experiments at 35C, experiments here are again conducted at “room temperature”. I appreciate the added nuisance of elevating the temperature and RT might be justified here based on those previous experiments. But the authors might want to consider elevated temps again in the future. As they know, endocytosis has been shown to use different pathways at 35C than RT. Furthermore, temperature does influence membrane fluidity which alters the mobility of membrane proteins.

- Cacna1aCitrine data: I find this a very interesting aspect of the paper, especially since it is the only place where the distinction between stalk-sites and swelling-sites is based on data rather than inference. But I was disappointed that there are practically no data for ‘simple’ calyces. This is one of the places where I would love to see just a few more experiments. If ‘very-simple’ calyces were non-existent in these mice, that would be very interesting!

- The data in Figure 5&6 is missing appropriate controls. One excuse for the lack of controls might be that the others have done both experiments before with very similar outcomes. Yet even in the original septin paper, there was no solution or antibody control. There is not even an indication of the specificity of the antibody and the vendor was not known for stringent quality control.

Reviewer #3 (Remarks to the Author):

The manuscript of Adam Fekete et al. studied the mechanisms underlying the heterogeneity of synaptic transmission at mature calyx synapses using a combination of methods, including genetics, imaging, electrophysiology and modeling. The results suggest that two morphological modules, swelling and stalks, have distinct topology of Ca²⁺ channel clusters and spatial coupling distance to synaptic vesicles, which may account for different release probability and short-term plasticity in calyces with different morphology. The experiments are very difficult and challenging. The authors have performed these experiments elegantly and extensively. The conclusions are important. I have only some suggestions that may improve the manuscript before publication.

1. In experiments, authors used EGTA, SP20, and simulation to estimate the distance between Ca²⁺ channels and vesicles. This is indirect. The authors may want to admit this point and soften their conclusions.
2. Large EPSCs may reach up to 20 nA, which may cause saturation of postsynaptic receptors. This might in part account for the smaller effects of EGTA effects on Simple and complex calyces (Fig. 5). The authors may want to repeat some experiments with competitive AMPA receptor blockers to relieve the saturation and desensitization of postsynaptic glutamatergic receptors.
3. In Fig4., authors calculated cluster volume, one of central features of this manuscript. I have not been able to find the method of calculating cluster volume. Fig. 4g-i and Fig1.b are almost identical, except the fit. The explanation for the curve fit can be in more detail.
4. Authors should discuss more about the swelling and stalk modules in the introduction. Can the two modules be switched back and forth as the terminal matures? If two modules are found beyond the calyx of Held synapses, it may also broaden the impact of the current study.
5. Sheng et al. (Ref. 25) performed recordings of calcium channel currents, vesicle release, and release probability at single active zones of the calyx. They found that single active zones contained 5 – 218 (mean: 42) calcium channels and 1 – 10 readily releasable vesicles (RRVs). Variation of the calcium channel number caused variation of the RRV's release probability (PRRV) and RRV's number, resulting in heterogeneous release properties among different active zones. By controlling PRRV, the number of calcium channels determined whether release is facilitated or depressed during repetitive stimuli. They suggest that regulation of the calcium channel density is a major mechanism to yield synapses with different release properties and short-term plasticity. The present work demonstrates a physiological example indicating that calcium channel number is crucial in deciding release probabilities in two modules of the calyces. The authors may want to discuss the results of Sheng et al. (Ref 25) in more detail to strengthen their conclusions in Introduction and Discussion.

Page 18, discussion, 2nd paragraph, line 1: The number of VGCCs in the cluster seems to be crucial factor of Pr as they correlate with the active zone area in hippocampal synapses (16). The large variability in the number of VGCCs per cluster implicate that the cluster size diversified Pr at individual active zones (26).

The above discussion did not cite Ref. 25. Unlike Ref 16 or 26 quoted that relied on simulation to make a suggestion, Ref 25 shows at the same active zone that the number of VGCCs (recorded via current recordings) determines the release probability of the release-ready vesicles (recorded with capacitance measurements). A discussion of the results of Ref 25 may strengthen this Discussion section.

6. In Fig.1a, each swelling and stalk should be indicated with an arrow in the figure.
7. In Fig.1a, ($20 < \#$ of Swellings) should be ($\#$ of Swellings > 20).
8. Fig.2-3 showed the Ca^{2+} transient difference between stalk and swelling contributes to the difference of Ca^{2+} current in complex and simple Calyxes. Combining these two figures might be better.
9. In Fig.7, authors should show the simulation equation(s) used in fig.7.

General Response to all Reviewers:

We would like to thank all three referees for their constructive comments on this manuscript in the previous round of review. Having carefully digested these criticisms, we substantially reorganized the paper in text and figure presentation order with the goal to improve its logic flow, clarity of interpretation and text readability.

We would like to emphasize that the novelty of our conceptual model for this study is multifaceted. **(1)** Our results for the first time demonstrated that central nerve terminals vary morphological complexity, i.e. the ensembles of two distinct morphological modules (high Pr stalks vs low Pr swellings), to endow a homogenous population of synapses with heterogeneous outputs. **(2)** This is achieved by varying bimodal organization of Ca²⁺ channel clusters relative to SVs in two distinct modules with septin filaments being the key molecular substrate to differentiate the coupling distance. **(3)** The nerve terminal can reciprocally tune Pr in each module, diversifying the summated global Pr along the morpho-functional continuum. This simple biological solution exemplifies how neuronal circuits expand capacity for information coding and filtering, linking synaptic heterogeneity to physiological functionality.

Our main conclusion on modular ensembles for synaptic heterogeneity is not only built on our results included in this study, but also previous studies using freeze-fracture labeling of VGCC clusters in Nakamura et al. (2015). We have re-examined the published datasets of VGCC clusters, and noted that the distribution pattern of VGCCs is not unimodal, and instead best described by bimodal Gaussian fits in P14/21 calyces (as illustrated in the figure below). Unlike our study in which we map each cluster onto different compartments of the same terminal, freeze-fracture occurs randomly, precluding precise localization of VGCC clusters in the calyceal terminal. This result nevertheless support bimodal cluster organization of VGCCs, in line with our results showing large clusters in stalks and small clusters in swellings.

Fit with Gaussian

A detail account of our changes is outlined by **blue letters in the revised manuscript and below in our point-to-point rebuttal** in which we believe we have fully resolved all issues of concern.

Reviewer #1 (Remarks to the Author):

The present MS describes experiments aiming to understand the mechanisms underlying the heterogeneity of functional properties among Calyx of Held synapses. The authors built on their previous finding that calyx synapses show substantial diversity and here they performed a series of physiology, modelling and anatomy experiments to demonstrate that different coupling distances underlie different release probabilities and that the functional diversity correlates with the structural heterogeneity of these synapses. The reviewer found the concept of creating heterogeneity by two distinct mechanisms and blending them in a variable degree interesting and novel. Despite this, there are few issues with the MS, which should be addressed.

- 1. The experiments with high-affinity Ca²⁺ indicators do not add anything to the MS. The authors themselves state that the differences in Pr cannot be explained by global Ca events, because that depends on the volume of the structures. Then the authors performed Ca²⁺ measurements with a low-affinity dye (Fluo4FF) to test nanodomain Ca²⁺ concentrations. However, for some reasons the authors used long trains of APs, the interpretation of which is very complicated given its dependence on diffusion, Ca buffering and potentially even extrusion. It is also unknown why the authors decided to illustrate this potentially more meaningful finding in the supplementary figure.*

Response:

We appreciate the Reviewer's comment and agree with the Reviewer that Ca²⁺ imaging results are not intuitive to explain differences in Pr between stalks and swellings. We have now moved Ca²⁺ imaging dataset with high-affinity Fluo-4 to Supplementary Figure 3 to simply support the conclusion that action potentials reliably propagate into all compartments of the calyx and evoke Ca²⁺ influx. We have also moved Ca²⁺ imaging data with low-affinity Fluo-4FF to the main text (Figure 8) to provide mechanistic explanations for expanded RRP size and spike-firing fidelity in complex calyces with increasing number of swellings.

We amended the text in Results (Page 8 and 15) and Discussion (Page 18).

- 2. Despite their own argument for the lack of meaningful interpretation of the Ca transients in swellings and stalks, the authors start the next paragraph (page 9) that 'As we found that swellings are characterized by larger single AP-evoked Ca²⁺ transients than stalks, we reasoned*

*that the total number and the density of VGCCs may be higher...". This does not make sense!
Only differences in the surface to volume ratio can explain the data.*

Response:

We have revised the rationale for performing Ca²⁺ imaging experiments and interpretations of results with clear acknowledgement of surface/volume ratio as the main contributor to the larger Ca²⁺ transients in swellings than stalks.

We amended the text in Results (Page 8 and 9).

- 3. The authors describe their measurements on cluster geometry as volume (3D structure). The reviewer is sceptical about the resolution of their technique to provide reliable estimate of the real cluster volume. It is also interesting why the authors decided to measure the 'volume' and not the 2D projection surface area? That would be much closer to the relevant morphological structure! The authors should definitely provide some evidence that their measure has anything to do with the AZ geometry!*

Response:

We agree with the Reviewer, information about the size of the clusters is lost below the resolution limit of the microscope (~120 nm laterally with Huygens deconvolution; voxel size, x,y pixel spacing = 51.2 ± 2.0 nm, z-axis step = 300 nm), and thus the technique does not provide strict estimate of the real geometry (volume, surface area) to directly relate the measures to AZ geometry (diameter, area). To address the raised issues, we quantified the cumulative citrine fluorescence intensity per clusters (or the sum of fluorescence intensities of all voxels in the cluster) on the two morphological modules using the 3D object counter plugin in ImageJ. Cumulative citrine intensity per clusters is a direct correlate of the VGCC number per cluster because the Cav2.1 Ca²⁺ channels were genetically tagged with citrine ensuring 100% labeling efficiency (N-terminus of α_1 -subunit, Cacna1a^{citrine} mouse line; Mark et al., 2011). We represented the data as normalized values (Fig. 4f) because the fluorescence intensities were not calibrated but paired between clusters in swelling and stalk located in equal depths from the slice surface. Furthermore, we expanded the description of the image analyses both in the Results and Methods section. Using these comparative values, we detected 34.7% larger clusters on stalks than swellings. It is unlikely that neighboring VGCC clusters are identified as single cluster (see the separation of peaks in the profile plot in Fig. 4b) because we have shown that one cluster associates to one AZ and

the nearest neighbor distance between VGCC clusters is larger than ~780nm, a separation well above the resolution limit (Taschenberger et al., 2002; Nakamura et al., 2015).

We amended the text in Results (Page 10), Discussion (Page 19), Methods (Page 31) and Figure Legends (Fig. 4b,f; Page 48)

4. *The EGTA experiments indicate different coupling distances among Calyces with different number of swellings. The reviewer is puzzled about the top two sentences in page 13! The authors state that a shorter coupling distance should have shorter synaptic delay and EPSC time course, but their data is inconsistent with this. What does it mean?*

Response:

We did not see a difference in synaptic delays (SD) between simple and complex calyces likely because they both contain the stalk module with tightly coupled SVs at 14 nm distance which fuse first after VGCC opening and thus control the initial EPSC rise time and SDs. Although we could not quantify the EGTA effect on SD in these experiments because we removed the pipette from the terminal after EGTA injection, previously we found no effect of EGTA in mature calyces (Nakamura et al., 2015). Consistent with the presence of tightly coupled SVs in the stalk module, (a), the miniature and evoked EPSC time courses were comparable; (b), the EPSC time courses of simple and complex calyces were not different; and (c), EGTA injection did not affect the EPSC time course differentially in simple and complex calyces. Our data recapitulate the slight inhibitory effect of 10 mM EGTA on the time course of EPSCs observed by Nakamura et al. (2015).

We amended the text in Results (Page 12) regarding synaptic delays and time course of EPSCs.

5. It is much less informative to provide the ratios of two values (e.g. in fig. 5B) than providing both values in a pairwise plot. It is also true for the result section where the authors describe the effect of EGTA; they should tell that the value was reduced from X to Y rather than telling their ratios. And in few other places in the MS.

Response:

Corrected.

Reviewer #2 (Remarks to the Author):

The work by Fekete et al presents a large number of technically very difficult experiments that together strongly support the idea that release probability at the Calyx of Held is regulated by synaptic morphology, by the distribution of calcium channel and by the proximity of synaptic vesicles to calcium channels. At the Calyx of Held all vesicle release sites are formed by the very same presynaptic axon, making a particularly attractive target for studies on release site diversity. The authors describe two morphological ‘modules’: one with vesicles in close proximity to large calcium channel clusters and a high release probability; the other with fewer calcium channels, less tight coupling and lower release probability. They propose that the overall characteristics of a Calyx is determined by varying amounts of these two modules.

- 1. Overall, the data is convincing (see below) and while the writing could be clearer the authors explain the context and background of the experiments well. A bit more discussion on the physiological relevance of fine-tuning at this particular synapse would be helpful.*

Response:

We agree with the Reviewer and have reorganized the entire paper and revised the text wording to improve the readability of this paper. We have also expanded the Discussion with the physiological relevance of fine-tuning at the mature calyx of Held synapse.

We amended the Discussion (Page 22).

- 2. The major conclusions of the paper are not very novel and are hardly surprising. (I appreciate that I might have missed a crucial part in which case the writing is not as clear as I thought!) Very similar experiments and results have been published before by the same group and others. Confirmation is always good, especially when experiments are as difficult as these! One value of the present publication then lies in being a comprehensive compendium and as such will serve as a valuable one-stop reference. That said, the use of a citrin-P/Q knock-in to determine calcium channel density and spatial clustering is clever and very valuable.*

Response:

We acknowledge the Reviewer's comment on the novelty issue. We would however argue that our work represents a major conceptual advancement in our understanding of the origin of synaptic diversity for several compelling reasons.

1. We discovered that the morpho-functional diversity arises from different ensembles of two **distinct morphological modules**, i.e. large and small presynaptic compartments. The large ones are comprised of **large clusters of Ca²⁺ channels tightly coupled** to synaptic vesicles (SVs) with high release probability (Pr) while the small ones contain **small clusters of Ca²⁺ channels loosely coupled to SVs** with low release Pr (i.e. Tight- vs. Loose-Nanodomain Modules).
2. We demonstrate that the **septin** filaments are differentially engaged in regulating the coupling distance of Ca²⁺ channels to SVs in the two distinct morphological modules. This is the first evidence that a specific cytomatrix protein plays a role in creating synaptic heterogeneity.
3. We generate the first 3D map of Ca²⁺ channel clusters in any central terminals using an innovative approach for visualizing these channels that are notoriously difficult to delineate, and conceptualize a **framework for subsynaptic organization of Ca²⁺ channels** in different modules.
4. Using **computational modeling**, we quantitatively simulated the effects of Ca²⁺ channel cluster and coupling distance as core determinants of Pr and fully recapitulate the synaptic heterogeneity as a function of morphological continuum.
5. Finally, we present surprising evidence that the fidelity and timing of **postsynaptic spiking** during high-frequency trains is critically **dependent on these presynaptic modules**.

With these being said, we made every effort to highlight these novel findings throughout this revised version.

3. *The idea of two modules is interesting but clashes with the concept of the 'morpho-functional continuum' that the authors keep referring to. In fact, a duality only arises when the authors create two distinct and arbitrary sets (less than 10 'swellings' and >20 swellings; including the 10-20 data as a third set was previously done by the same others). I suggest that the authors more clearly formulate whether they want to treat this as a two-module or a continuum idea for this publication.*

Response:

We appreciate this comment and revised the text accordingly to indicate the term “module” refers to the morphological subunits (swelling and stalk) of the presynaptic terminal as functional subunits. In compliance with the Reviewer’s suggestion, we have now clearly formulate the concept that the continuum is composed of varying ensembles of these two functional modules in which the clusters of VGCCs and their coupling distance are different.

We amended the text to avoid inconsistent use of the term module in Introduction (Page 4 and 5) Results (e.g. Page 17) and Discussion (e.g. Page 20).

4. *More detailed comments:*

• *One of my favorite parts of the paper is shown in Fig 4i. It took me a while to appreciate what was fit and what was derived, but the way I got to understand it was that the straight line in 4i is based on fitting data in 4g and 4h (grey) and then superimposing the result on to the data in 4i. This could probably be made a lot clearer than it currently is. Also, that the data in 4g-l is the same as in Fig1b must be indicated much more clearly. I also would have liked to see the straight line fit from 1b superimposed onto the model fit in 4i. But the fact that after so many divisions, normalizations and extrapolations the model superimposes onto the data is remarkable!*

Response:

We agree with the Reviewer that we need to increase clarity and decrease redundancy between figures as much as possible. Hence, we deleted the RRP data from Fig. 4 (identical to Fig. 1b RRP scatter plot) and clearly stated in the figure legends that the Fig. 4g,h scatter plots are identical to Fig. 1b but the fitting ($\pm 95\%$ confidence band; Fig. 4g) or prediction ($\pm 95\%$ prediction band; Fig. 4h) are different. We also superimposed the linear fit of synaptic strength from Fig. 1b on Fig. 4h for comparison (blue trace).

We amended Results (Page 11), Fig. 4g,h and legends (Page 48).

• *Figure 4i also slightly inoculates against one of my main concerns about the data and how it is presented. Much of the data is derived from calculations, extrapolations and fits, all of which have variance associated with them. None of this variance is carried forward in any of the calculations or data presentations, leaving the reader at a loss of how well the data conforms to all the models and assumptions. In other words, how robust are the conclusions? For example, the decrement in amplitude during a train is fit with two exponentials and the RRP is calculated from back-extrapolation to the first stimulus. As has been pointed out many times, this is clearly in error as it does not take replenishment of*

pools into account. Moreover, Figure 1a, complex shows amplitudes that first decrement to a low level, but then increase again by at least 50% for the last 100 ms. Back-extrapolating from the last 50ms thus clearly is not appropriate. Would a more complex model make a difference to the overall conclusion? I would think so, but it also might introduce yet another variable making sound conclusions and predictions even harder.

Response:

We agree with the Reviewer and have added 95% confidence intervals to the fittings. We have carefully considered all propagated errors for the calculations and come to the conclusion that **these errors are too small to be meaningful to the functional diversity of the mature calyceal terminals.**

1. Variation of derived data. In order to demonstrate the variation of derived data we determined the \pm 95% confidence band for fittings (Fig. 1b, 3b,c, 4d,g, 5b,c, 6c, 9a), the \pm 95% prediction band for calculation (Fig. 4h), and the effect of \pm standard deviation (SD) of a previously derived fit parameter on a subsequently fitted mean curve (see modified Fig. 4g and 9a below). Red lines represent variation in fitted mean curve when \pm SD (68%) of previously derived fit parameters are considered. Variation in the numbers of VGCC clusters per swellings and on stalks derived by linear fitting (Fig. 4d) is considered in Fig. 4g (see red traces on the left). Variation in the RRP per swellings and on stalks derived by linear fitting (Fig. 6c) is considered in Fig. 9a (see red traces on the right). The variations of the mean fits (<4%) are negligible.

We amended the text in Methods (Page 28).

2. Steady-state EPSC amplitudes (EPSCss). The Reviewer observed an elevation in the EPSCss of complex calyces affecting the last 50 ms of the 200 ms long 300 Hz train. This elevation was present in some but not all of the calyces either simple or complex (complex, 106 ± 33.82 pA, $n = 18$, simple, 78.2 ± 29.6 pA, $n=14$, $p = 0.5538$, $df = 30$). The elevation is a minor fraction of the total synaptic output (initial EPSCs are in the range of 7.6-23 nA), and signals a slight activation of Ca^{2+} -dependent replenishment of SVs because 10 mM EGTA injection eliminates it (Wang and Kaczmarek, 1998; paragraph 1 on page 1137 in Taschenberger et al., 2002). To determine the error in RRP and Pr (see modified Fig. 1b and 4g below) we fitted the control cumulative EPSC after RRP depletion but before EPSCss elevation at the 100-150 ms segment (red triangles, red mean traces) instead of the last 50 ms (black triangle, black mean traces \pm 95% confidence interval). The error was a marginal \sim 3% decrease in RRP and \sim 3-4% increase in Pr affecting simple (RRP, 2.8 %; Pr, 3.1 %) and complex calyces equally (RRP, 3.1 %; Pr, 3.7 %).

We amended the text in Methods (Page 28).

3. Replenishment of the RRP. The slope of the cumulative EPSC at steady-state represents the replenishment of SV pools. Thus the linear fitting method used for RRP size quantification takes the replenishment into account (Schneppenburger et al., 1999). The non-linear replenishment model of Wesseling and Lo (2002) considers a variable rate at which SVs are recruited during the train. We also quantified the size of RRP and Pr using the Wesseling and Lo (2002) model and found the size of RRP is elevated (simple, 4%; complex, 8.8%) and Pr (simple, 2.4%; complex, 7.6%) is depressed compared to the linear fitting, however, the heterogeneity among calyces is comparable with the two methods (see data included in Fig. 1b). Furthermore, in our previous study we also estimated RRP by the non-linear method, and the heterogeneity was comparable to the linear (Fig. 6A in Grande and Wang, 2011).

	Complex	Simple	p, df	Stat
RRP, Schneggenburger et al., 1999	91.8 ± 5.31 nA	53.5 ± 2.84 nA	< 0.0001, 25	u.t.t.W.
RRP, Wesseling and Lo, 2002	99.9 ± 5.98 nA	55.6 ± 3.47 nA	< 0.0001, 26	u.t.t.W.
Pr, Schneggenburger et al., 1999	0.157 ± 0.008	0.212 ± 0.012	0.0004, 30	u.t.t.
Pr, Wesseling and Lo, 2002	0.145 ± 0.008	0.207 ± 0.014	0.0009, 21	u.t.t.W.

We amended the text in Results (Page 6), Methods (Page 27) and Figure Legends (Fig. 1b; Page 46).

4. EPSC decay. As the Reviewer pointed out the decay of EPSC trains consist of more than two components including the depletions of two RRP subpools and replenishment of SVs. However, we have empirically found that the two-exponential function describes the decay time course well. In those cases where the EPSCs elevation during the last 50 ms of the 200 ms long train visibly affected the quality of fitting, we fitted the two-exponential function to a shorter segment (after facilitation but before the elevation).

We amended the text in Methods (Page 27).

- *While the authors have previously done experiments at 35C, experiments here are again conducted at “room temperature”. I appreciate the added nuisance of elevating the temperature and RT might be justified here based on those previous experiments. But the authors might want to consider elevated temps again in the future. As they know, endocytosis has been shown to use different pathways at 35C than RT. Furthermore, temperature does influence membrane fluidity which alters the mobility of membrane proteins.*

Response:

We agree with the Reviewer’s comment that the synaptic functions are influenced by the temperature in which slice recordings are conducted and thus we cannot directly translate our results measured at room temperature into *in vivo* physiological conditions. The temperature and consequently the membrane fluidity differences, however, are unlikely responsible for the heterogeneity in synaptic transmission at the calyx of Held synapse because we always performed the experiments under the same experimental conditions along the morpho-functional continuum. We have previously shown that such heterogeneity is apparent at 35⁰C except that the frequency-dependence of morpho-functional continuum is shifted (Grande and Wang, 2011).

We amended the text in Discussion (Page 22) in order to clarify and emphasize the limitations of our experiments and their interpretations.

• *Cacna1a^{Citrine} data: I find this a very interesting aspect of the paper, especially since it is the only place where the distinction between stalk-sites and swelling-sites is based on data rather than inference. But I was disappointed that there are practically no data for ‘simple’ calyces. This is one of the places where I would love to see just a few more experiments. If ‘very-simple’ calyces were non-existent in these mice, that would be very interesting!*

Response:

As per the Reviewer’s request we increased the number of simple calyces in Fig. 4d (from n = 6 to 10). We concluded that simple calyces exist in these mice (number of swellings ≤ 10 ; *Cacna1a^{Citrine}* line, C57 background).

We amended the text in Results (Page 10), Fig. 4d, and the text in the Figure Legend (Page 48).

• *The data in Figure 5&6 is missing appropriate controls. One excuse for the lack of controls might be that the others have done both experiments before with very similar outcomes. Yet even in the original septin paper, there was no solution or antibody control. There is not even an indication of the specificity of the antibody and the vendor was not known for stringent quality control.*

Response:

In these experiments we tested the effect of septin5 antibody (SP20, Santa Cruz, sc-65512) on the entire range of calyx complexity (from simple to complex calyces) establishing an internal control situation (SP20/Control) in which we could compare the antibody effect in simple versus complex calyces. Furthermore, instead of a solution or antibody control for these experiments we have performed even stronger set of control experiments in our earlier publication (see Fig. 6A-C in Yang et al., 2010). In these experiments we tested the specificity and potency of SP20 antibody by injecting it into CHO (Chinese hamster ovary) cells expressing Sept5-GFP or actin-GFP (1:200-1:2000) and into the calyx of Held synapses prepared from septin5 wild-type (*Sept5^{+/+}*) and septin5 KO (*Sept5^{-/-}*) mice (1:2000). We found large effect of SP20 on the EPSC amplitude in the *Sept5^{+/+}* mice (40.3 ± 7.9 % increase) but no effect at all on the EPSC amplitude in the *Sept^{-/-}* mice (1.0 ± 3.5 %) demonstrating that the SP20 antibody is a specific and potent functional inhibitor of Sept5 filaments in the calyx of Held synapses.

We amended the text in Methods (**Page 25**) to indicate specificity and potency of SP20 antibody.

Reviewer #3 (Remarks to the Author):

The manuscript of Adam Fekete et al. studied the mechanisms underlying the heterogeneity of synaptic transmission at mature calyx synapses using a combination of methods, including genetics, imaging, electrophysiology and modeling. The results suggest that two morphological modules, swelling and stalks, have distinct topology of Ca²⁺ channel clusters and spatial coupling distance to synaptic vesicles, which may account for different release probability and short-term plasticity in calyces with different morphology. The experiments are very difficult and challenging. The authors have performed these experiments elegantly and extensively. The conclusions are important. I have only some suggestions that may improve the manuscript before publication.

1. In experiments, authors used EGTA, SP20, and simulation to estimate the distance between Ca²⁺ channels and vesicles. This is indirect. The authors may want to admit this point and soften their conclusions.

Response:

We agree with the Reviewer that our independent approaches (EGTA, SP20, numerical simulation) indirectly infer the VGCC-SV coupling distance, and acknowledge that currently these indirect techniques are the only ways to test the distance that functionally couples the VGCCs to SVs, and future experiments to directly measure the distance are highly desirable. We also would like to emphasize that measurements of the width of APs in the morphological modules (Fig. 2) were necessary because variation in the local AP width and thus the local Ca²⁺ influx could be a source of variation in the EGTA effect on synaptic transmission (Nakamura et al., 2018).

Nevertheless, we appreciate the Reviewer's comments and amended the text in Discussion (Page 20).

2. Large EPSCs may reach up to 20 nA, which may cause saturation of postsynaptic receptors. This might in part account for the smaller effects of EGTA effects on Simple and complex calyces (Fig. 5). The authors may want to repeat some experiments with competitive AMPA receptor blockers to relieve the saturation and desensitization of postsynaptic glutamatergic receptors.

Response:

We compared the inhibitory effect of 10 mM EGTA on the amplitude of 1st EPSC in the absence (control) and presence of 1 mM kynurenic acid, and found no statistical significant difference between the two

conditions. Thus we conclude that saturation of postsynaptic AMPARs does not substantially affect the EGTA effect.

N = 9 for control and N = 6 for kyn, P = 0.17 in t-test

We have clarified this by citing previous publications on this issue (Joshi and Wang, 2002; Taschenberger et al., 2002; Ishikawa et al., 2002; Yamashita et al., 2003; Renden et al., 2005; Yang et al., 2011) to rationalize our experimental conditions (Page 7 and 14).

3. In Fig4., authors calculated cluster volume, one of central features of this manuscript. I have not been able to find the method of calculating cluster volume. Fig. 4g-i and Fig1.b are almost identical, except the fit. The explanation for the curve fit can be in more detail.

Response:

Our approach technically precludes us from providing strict estimate of the real volume due to the resolution limit of the microscope. To address the raised issues, we quantified the cumulative citrine fluorescence intensity per clusters (or the sum of fluorescence intensities of all voxels in the cluster) on the two morphological modules using the 3D object counter plugin in ImageJ. Cumulative citrine intensity per clusters is a direct correlate of the VGCC number per cluster because the Cav2.1 Ca²⁺ channels were genetically tagged with citrine ensuring 100% labeling efficiency (N-terminus of $\alpha 1$ -subunit, *Cacna1a*^{citrine} mouse line; Mark et al., 2011). We represented the data as normalized values (Fig. 4f) because the fluorescence intensities were not calibrated but paired between clusters in swellings and stalks located in equal depths from the slice surface. Furthermore, we expanded the description of the image analyses both in the Results and Methods section.

We amended the text in Results (Page 10), Discussion (Page 19), Methods (Page 31) and Figure Legends (Fig. 4b,f; Page 48)

We agree with the Reviewer that we need to decrease redundancy between figures as much as possible. Hence, we deleted the RRP data from Fig. 4 (identical to Fig. 1b RRP scatter plot) and clearly stated in the figure legends that the Fig. 4g-h scatter plots are identical to Fig. 1b but the fitting ($\pm 95\%$ confidence band; Fig. 4g) or prediction ($\pm 95\%$ prediction band; Fig. 4h) are different. We also superimposed the linear fit of synaptic strength from Fig. 1b on Fig. 4h for comparison (blue trace).

We amended Results (Page 11), Fig. 4g,h and legends (Page 48).

4. Authors should discuss more about the swelling and stalk modules in the introduction. Can the two modules be switched back and forth as the terminal matures? If two modules are found beyond the calyx of Held synapses, it may also broaden the impact of the current study.

Response:

We agree with the Reviewer that the swelling and stalk modules, and their plasticity, were not thoroughly discussed and corrected this caveat in the Introduction.

Our findings can be generalized to any neural circuits in which the postsynaptic neurons are innervated by multiple inputs with heterogeneous Pr-s from the same or diverse presynaptic neurons. The final output of postsynaptic neurons depends on the weighted average of Pr-s at release sites. This is in parallel to the integration at calyx, where the presynaptic neuron condenses inputs onto the soma of postsynaptic neuron to enhance the fidelity of spike timing and number, both of which are important cues for computation. It is tempting to speculate that the structure and function of other large, multi-compartmentalized nerve endings, e.g. cerebellar mossy terminal (CMT), are similarly organized. CMTs also contain 100s of AZs and are ultrafast. The impact of the presynaptic side may be considered similarly as the weighted mean of Pr-s. However, the information flow through the cerebellar mossy terminals is very divergent suggesting that the impact of a single CMT needs to be assessed at the level of a neural network.

We amended the text in Introduction (Page 4).

5. Sheng et al. (Ref. 25) performed recordings of calcium channel currents, vesicle release, and release probability at single active zones of the calyx. They found that single active zones contained 5 – 218 (mean: 42) calcium channels and 1 – 10 readily releasable vesicles (RRVs). Variation of the calcium channel number caused variation of the RRV's release probability (PRRV) and RRV's number, resulting in heterogeneous release properties among different active zones. By controlling PRRV, the

number of calcium channels determined whether release is facilitated or depressed during repetitive stimuli. They suggest that regulation of the calcium channel density is a major mechanism to yield synapses with different release properties and short-term plasticity. The present work demonstrates a physiological example indicating that calcium channel number is crucial in deciding release probabilities in two modules of the calyces. The authors may want to discuss the results of Sheng et al. (Ref 25) in more detail to strengthen their conclusions in Introduction and Discussion.

Response:

We agree with the Reviewer that our study provides an example how the variation in the number of VGCC at single AZs can diversify the function of an entire terminal and hence the clear citation of the findings of Sheng et al. (2012) are crucial for the context of our study.

We amended the text in Introduction (Page 5) and Discussion (Page 18).

6. Page 18, discussion, 2nd paragraph, line 1: The number of VGCCs in the cluster seems to be crucial factor of Pr as they correlate with the active zone area in hippocampal synapses (16). The large variability in the number of VGCCs per cluster implicate that the cluster size diversified Pr at individual active zones (26). The above discussion did not cite Ref. 25. Unlike Ref 16 or 26 quoted that relied on simulation to make a suggestion, Ref 25 shows at the same active zone that the number of VGCCs (recorded via current recordings) determines the release probability of the release-ready vesicles (recorded with capacitance measurements). A discussion of the results of Ref 25 may strengthen this Discussion section.

Response:

We thank to the Reviewer for pointing at this caveat.

We amended the text in Discussion (Page 18).

6. In Fig.1a, each swelling and stalk should be indicated with an arrow in the figure.

Response:

Corrected. We indicated the stalks with white triangles and the swellings with red crosses.

7. In Fig.1a, (20<# of Swellings) should be (# of Swellings > 20).

Response:

Corrected.

8. Fig.2-3 showed the Ca²⁺ transient difference between stalk and swelling contributes to the difference of Ca²⁺ current in complex and simple Calyxes. Combining these two figures might be better.

Response:

We considered this and other Reviewers' comments and decided to re-organize the text and figure presentation to improve the logic flow of storyline, as well as to adhere to the editorial requirements for the length of figure legends (<350 words) and the number of figures (< 10).

9. In Fig.7, authors should show the simulation equation(s) used in fig.7.

Response:

Corrected.

REFERENCES

1. Grande, G. & Wang, L.Y. Morphological and functional continuum underlying heterogeneity in the spiking fidelity at the calyx of Held synapse in vitro. *J. Neurosci.* **31**, 13386-13399 (2011).
2. Ishikawa, T., Sahara, Y. & Takahashi, T. A single packet of transmitter does not saturate postsynaptic glutamate receptors. *Neuron* **34**, 613-621 (2002).
3. Joshi, I. & Wang, L.Y. Developmental profiles of glutamate receptors and synaptic transmission at a single synapse in the mouse auditory brainstem. *J. Physiol.* **540**, 861-873 (2002).
4. Mark, M.D., Maejima, T., Kuckelsberg, D., Yoo, J.W., Hyde, R.A., Shah, V., Gutierrez, D., Moreno, R.L., Kruse, W., Noebels, J.L. & Herlitze, S. Delayed postnatal loss of P/Q-type calcium channels recapitulates the absence epilepsy, dyskinesia, and ataxia phenotypes of genomic Cacna1a mutations. *J. Neurosci.* **31**, 4311-4326 (2011).
5. Nakamura, Y., Reva, M. & DiGregorio, D.A. Variations in Ca²⁺ Influx Can Alter Chelator-Based Estimates of Ca²⁺ Channel-Synaptic Vesicle Coupling Distance. *J. Neurosci.* **38**, 3971-3987 (2018).
6. Nakamura, Y., Harada, H., Kamasawa, N., Matsui, K., Rothman, J.S., Shigemoto, R., Silver, R.A., DiGregorio, D.A. & Takahashi, T. Nanoscale distribution of presynaptic Ca⁽²⁺⁾ channels and its impact on vesicular release during development. *Neuron* **85**, 145-158 (2015).
7. Renden, R., Taschenberger, H., Puente, N., Rusakov, D.A., Duvoisin, R., Wang, L.Y., Lehre, K.P. & von Gersdorff H. Glutamate transporter studies reveal the pruning of metabotropic glutamate receptors and absence of AMPA receptor desensitization at mature calyx of Held synapses. *J. Neurosci.* **25**, 8482-8497 (2005).
8. Schneggenburger, R., Meyer, A.C. & Neher, E. Released fraction and total size of a pool of immediately available transmitter quanta at a calyx synapse. *Neuron* **23**, 399-409 (1999).
9. Sheng, J., He, L., Zheng, H., Xue, L., Luo, F., Shin, W., Sun, T., Kuner, T., Yue, D.T. & Wu, L.G. Calcium-channel number critically influences synaptic strength and plasticity at the active zone. *Nat. Neurosci.* **15**, 998–1006 (2012).
10. Taschenberger, H., Leao, R.M., Rowland, K.C., Spirou, G.A. & von Gersdorff, H. Optimizing synaptic architecture and efficiency for high-frequency transmission. *Neuron* **36**, 1127–1143 (2002).
11. Wang, L.Y. & Kaczmarek, L.K. High-frequency firing helps replenish the readily releasable pool of synaptic vesicles. *Nature* **394**, 384–388 (1998).
12. Wesseling, J.F. & Lo D.C. Limit on the role of activity in controlling the release-ready supply of synaptic vesicles. *J. Neurosci.* **22**, 9708-9720 (2002).

13. Yamashita, T., Ishikawa, T. & Takahashi, T. Developmental increase in vesicular glutamate content does not cause saturation of AMPA receptors at the calyx of Held synapse. *J. Neurosci.* **23**, 3633-3638 (2003).
14. Yang, Y.M., Aitoubah, J., Lauer, A.M., Nuriya, M., Takamiya, K., Jia, Z., May, B.J., Huganir, R.L. & Wang, L.Y. GluA4 is indispensable for driving fast neurotransmission across a high-fidelity central synapse. *J. Physiol.* **589**, 4209-4227 (2011).
15. Yang, Y.M., Fedchyshyn, M.J., Grande, G., Aitoubah, J., Tsang, C.W., Xie, H., Ackerley, C.A., Trimble, W.S. & Wang, L.Y. Septins regulate developmental switching from microdomain to nanodomain coupling of Ca⁽²⁺⁾ influx to neurotransmitter release at a central synapse. *Neuron* **67**, 100–115 (2010).

REVIEWERS' COMMENTS:

Reviewer #1 (Remarks to the Author):

The authors have adequately addressed my main concerns and restructured the MS in a way that it is much better accessible to the readers.

The authors argument on EGTA effect on latency and time course is still not very clear.

VGCC-SV coupling distances can also influence the synaptic delay (SD; 31). We compared the delays in

229 simple and complex calyces (see SD1 and SD2 in Fig. 5a) and found no detectable difference

230 (Supplementary Fig. 5a and Supplementary Table 2). We attribute this to the omnipresent high Pr

231 release sites which dominate the first latency to release and thus preferentially influence the delay.

232 Consistent with this finding, (a), the miniature and evoked EPSC time courses were comparable;

Why should the time course of the EPSC necessarily reflect differences in delay? The reviewer suggests that the authors re-consider this argument and rephrase it in the MS.

Still some minor points:

The authors should not use the Y label of the graphs on top of the graph! Please use the conventional way of labelling the axis!

Fig 5E: are those (blue) EGTA or SP20 data?

Lines 120, 121: what are the n = 8/6 and 11/6?

Line 136: APs were elicited in current-clamp mode. Please phrase it precisely. CC mode does not elicit anything!

Line 172/173: please provide the variance! 509+-? and 709+-?

Line 183: please provide the variance!

Reviewer #2 (Remarks to the Author):

The authors have done a good job in addressing the concerns of this reviewer. The added text helps in making the paper more accessible. While I previously questioned the novelty, I do agree that the hypothesis of regulating synaptic function by employing distinct modules with different properties is noteworthy. And the fact that their data is so beautifully replicated by their model And the interpretation of the data given by the authors is in line with this hypothesis.

However, the data also perfectly in line with the alternative hypothesis, namely a gradual change of properties. I do not fully understand why the authors dig in their heels rather than allow for this.

They offer two points to suggest a bimodal rather than unimodal distribution:

1) Figure 5e (nb. should be SP20 not EGTA): SP20 has no effect on simple, but affects complex. While this seems to indicate duality, I bet that the increase in effect is gradual when calyces with intermediate swellings are considered. But his could indeed be the strongest argument (and controls would be important).

2) A figure in the rebuttal that fits two Gaussians to VGCC distributions. However, any of a large number of unimodal but skewed distributions would obviously fit just as well.

A third test could be the model. Assuming two populations, they can well describe the measured data. But they do not test the alternative.

Reviewer #3 (Remarks to the Author):

The authors have addressed all my concerns. The manuscript is ready for publication.

General response to all Reviewers:

We would like to sincerely thank the three reviewers and editors for their constructive comments on the manuscript. We outline our revisions by blue letters in the revised manuscript and the point-to-point rebuttal below.

Point-by point response to Reviewers' comments:

Reviewer #1 (Remarks to the Author):

The authors have adequately addressed my main concerns and restructured the MS in a way that it is much better accessible to the readers.

The authors argument on EGTA effect on latency and time course is still not very clear.

VGCC-SV coupling distances can also influence the synaptic delay (SD; 31). We compared the delays in simple and complex calyces (see SD1 and SD2 in Fig. 5a) and found no detectable difference (Supplementary Fig. 5a and Supplementary Table 2). We attribute this to the omnipresent high Pr release sites which dominate the first latency to release and thus preferentially influence the delay. Consistent with this finding, (a), the miniature and evoked EPSC time courses were comparable; Why should the time course of the EPSC necessarily reflect differences in delay? The reviewer suggests that the authors re-consider this argument and rephrase it in the MS.

Response: We have reworded these sentences to make it straightforward and amended the text on **page 11**.

Still some minor points:

The authors should not use the Y label of the graphs on top of the graph! Please use the conventional way of labelling the axis!

Response: We corrected all Y-axis labels as per the Reviewer's request.

Fig 5E: are those (blue) EGTA or SP20 data?

Response: Mislabeled is corrected (Fig 5E). The blue sets of data represent the SP20 experiments.

Lines 120, 121: what are the $n = 8/6$ and $11/6$?

Response: The meaning of these numbers is now explained in the text (**page 6-7**).

Line 136: APs were elicited in current-clamp mode. Please phrase it precisely. CC mode does not elicit anything!

Response: We amended the sentence in the text (**page 7**).

Line 172/173: please provide the variance! 509+-? and 709+-?

Response: We complemented the data with variance (page 9).

Line 183: please provide the variance!

Response: We complemented the data with variance (page 9).

Reviewer #2 (Remarks to the Author):

The authors have done a good job in addressing the concerns of this reviewer. The added text helps in making the paper more accessible. While I previously questioned the novelty, I do agree that the hypothesis of regulating synaptic function by employing distinct modules with different properties is noteworthy. And the fact that their data is so beautifully replicated by their model And the interpretation of the data given by the authors is in line with this hypothesis.

However, the data also perfectly in line with the alternative hypothesis, namely a gradual change of properties. I do not fully understand why the authors dig in their heels rather than allow for this.

They offer two points to suggest a bimodal rather than unimodal distribution:

1) Figure 5e (nb. should be SP20 not EGTA): SP20 has no effect on simple, but affects complex. While this seems to indicate duality, I bet that the increase in effect is gradual when calyces with intermediate swellings are considered. But his could indeed be the strongest argument (and controls would be important).

2) A figure in the rebuttal that fits two Gaussians to VGCC distributions. However, any of a large number of unimodal but skewed distributions would obviously fit just as well.

A third test could be the model. Assuming two populations, they can well describe the measured data. But they do not test the alternative.

Response: We acknowledge and appreciate the point of view from this Referee. Given the technical limitations that preclude us and others from accurately counting the number of calcium channels per cluster, we concede that other alternative models may work. We have removed the wording “bimodal” throughout the text to soften our arguments and conclusions.

Reviewer #3 (Remarks to the Author):

The authors have addressed all my concerns. The manuscript is ready for publication.

NA